# Single-cell mass spectrometry imaging of lipids in HeLa cells via tapping-mode scanning probe electrospray ionization

Yoichi Otsuka [1,2,3] ✉, Kazuya Kabayama [2,3,4], Ayane Miura[2], Masatomo Takahashi [5], Kosuke Hata [5], Yoshihiro Izumi [5], Takeshi Bamba [5], Koichi Fukase [2,3,4] & Michisato Toyoda [1,2,3]

The development of analytical technology that allows investigation of the diversity of cells that form biological tissues based on molecular information is important to elucidate the heterogeneity of cells and pathological mechanisms. Here, we present a proof-of-concept demonstration of single-cell mass spectrometry imaging (SC-MSI) via tapping-mode scanning probe electrospray ionization (t-SPESI), which is an atmospheric-pressure sampling ionization technique. We developed a novel t-SPESI unit that can be used in combination with an inverted fluorescence microscope and basic technologies to extract components from microregions of cells and measure ions with high sensitivity. We performed multimodal (fluorescence, lipid ion, and topographic) imaging of two types of HeLa cells labeled with fluorescent dyes and chemically fixed and showed the potential for subcellular-scale analysis of both cell structure and chemical composition. Furthermore, we evaluated the lipid species by comparing the SC-MSI results with those of supercritical fluid chromatography tandem mass spectrometry. The technical advancement presented here is effective for distinguishing cell types based on the signal intensity of lipid ions in single cells and investigating differences in the subcellular localization of lipids in different types of cells.

In living organisms, cells form functionally and structurally hierarchical tissues, and homeostasis is maintained via higher-order coordination of chemical reactions within and between cells. Local heterogeneity in the cellular networks that constitute tissues leads to disease such as malignant tumor[1], neurogenerative diseases[2] and fibrotic diseases[3]. Lipids, which are the major components of cell membranes, play diverse roles in intracellular and extracellular metabolic reactions, including in energy storage and as lipid mediators. Thus, abnormalities in lipid metabolism have been associated with disease[4–7].

Mass spectrometry is distinguished by its high sensitivity, wide dynamic range, and ability to identify molecular structures. Advances in lipidomics using liquid chromatography–mass spectrometry (LC–MS) have led to comprehensive investigations of lipids involved in diseases[8–10], and rare cells, such as cancer stem cells and circulating tumor cells, are clearly different types of cells than other cell populations[11,12]. However, the causal relationship between the heterogeneity of the cells constituting tissues and

the disease state remains unexplored. To elucidate the mechanism of malignant changes in disease and advance pathological diagnosis, techniques must be developed to analyze changes in the chemical state of individual cells in a diseased tissue based on diverse molecular information and identify groups of molecules involved in disease.

Mass spectrometry imaging (MSI) is a technique used to visualize the distribution of multiple components in biological tissues. The components in a local region of a tissue section are converted to gas-phase ions using several ionization techniques, and are measured with a mass spectrometer to obtain a mass spectrum. Information on the distribution of multiple species in a tissue can be obtained in a single measurement by combining the coordinate information of the ionized region of the sample with the ion signal intensities. Compared with imaging technologies such as optical microscopy, MSI has the advantage in that it can obtain multiple molecular-specific images in a single measurement. By reducing the area of the region subjected to ionization on the tissue, the cellular heterogeneity of diseased

[1]Department of Physics, Graduate School of Science, The University of Osaka, Toyonaka, Osaka, Japan. [2]Department of Chemistry, Graduate School of Science, The University of Osaka, Toyonaka, Osaka, Japan. [3]Forefront Research Center, Graduate School of Science, The University of Osaka, Toyonaka, Osaka, Japan. [4]Interdisciplinary Research Center for Radiation Sciences, Institute for Radiation Sciences, The University of Osaka, Toyonaka, Osaka, Japan. [5]Division of Metabolomics, Medical Research Center for High Depth Omics, Medical Institute of Bioregulation, Kyushu University, Fukuoka, Fukuoka, Japan. ✉e-mail: otsuka@phys.sci.osaka-u.ac.jp

tissues can be visualized based on the changes in metabolism within the cell. Since mammalian cells have a volume on the order of pL and a diameter of approximately 10 μm[13], highly accurate and sensitive ionization techniques are essential.

In recent years, technological advancements have been made to realize single-cell MSI (SC-MSI), with which the distribution of components in a single cell can be visualized with a high spatial resolution[14–20]. In secondary ion mass spectrometry (SIMS), primary ion beam is used to sputter a sample surface, and the secondary ions produced are measured. Depending on the ion beam focusing and secondary ion detection method, SIMS can be classified into nanoscale secondary ion mass spectrometry (NanoSIMS) and time-of-flight secondary ion mass spectrometry (TOF-SIMS)[21]. NanoSIMS allows visualization of intracellular elemental distributions in adipocytes[22] and stem cells[23] at pixel sizes ranging from 30 to 100 nm. Combined with isotope labeling, the distribution of drugs in local regions of intracellular structures has been visualized[24,25]. SC-MSI has been conducted via TOF-SIMS, and imaging of lipid fragments in glial cells[26] and cholesterol in macrophages[27] has been reported. SIMS has the advantage of allowing a subcellular resolution; however, high-energy primary ion beams can cause fragmentation of biomolecules, limiting the molecular information. The use of gas cluster ions reduces fragmentation and enables MSI with a pixel size of approximately 1 μm. SC-MSI of cardiomyocytes[28] (pixel size of 1.2 μm) using water cluster ion beams, MCF-7 cells[29] (pixel size of 0.2 μm) and breast cancer tissue[30] (pixel size of 1 μm) using argon cluster ion beams, and HeLa M cells using $C_{60}$ ion beams[31] (pixel size of 1 μm) has been reported.

In matrix-assisted laser desorption/ionization (MALDI), when a mixture of matrix and sample is irradiated with a focused ultraviolet pulsed laser beam, a portion of the photon energy is converted into kinetic energy and a desorption occurs. The matrix is also excited electronically to facilitate the ionization processes. SC-MSI has been achieved by reducing the laser spot size and improving the ionization efficiency via a post-ionization technique. MSI has been reported for HEK293 cells (pixel size of 1.5 μm)[32], HeLa cells (pixel size of 7 μm)[33], myeloma cells (pixel size of 5 μm)[34], mouse brains (pixel size of 1.4 μm)[35], RAW264.7 cells (pixel size of 5 μm)[36], Vero B4 cells (pixel size of 600 nm)[37], gastric carcinoma cells (pixel size of 10 μm)[38], Vero-B4 cells (pixel size of 2 μm)[39] and cocultured PANC-1 cells and PSC cells (pixel size of 10 μm)[40]. Although MALDI has the advantage of obtaining intact molecular ions, the matrix selection and coating methods to reduce the crystal size of the matrix and improve the spatial resolution of MSI are time consuming[41]. Other techniques have been developed to reduce the pixel size by focusing on laser light. SC-MSI of HeLa cells with a pixel size of 250 nm was achieved by using near-field light[42] and femtosecond UV laser light focused by a microlens fiber[43]. Vacuum ultraviolet laser desorption/ionization (VUVDI) allows visualization of metabolites and drugs in the plasma membrane, cytoplasm, and nucleus in three-dimensional space (voxel size of $300 \times 300 \times 25$ nm$^3$)[44]. Also, sub-micron order of spatial resolution images for putative endocannabinoids in single neurons of the leech segmental ganglion was reported using a projection-type imaging mass spectrometer[45].

Atmospheric-pressure sampling ionization (ASI) utilizing electrospray ionization (ESI)[46] has an advantage over MALDI and SIMS in that it requires minimal or no sample pretreatment and allows rapid MSI via soft extraction and ionization of analytes in an atmospheric-pressure environment. To realize SC-MSI via ASI, the extraction region must be miniaturized. Desorption electrospray ionization (DESI)[47] is a technique in which charged droplets are accelerated by high-velocity nitrogen gas flow and sprayed onto a sample surface to locally extract and ionize the sample components. Through optimization of the charged droplet sprayer, MSI of mouse brain tissue with a pixel size of 25 μm was achieved[48]. In nanospray desorption electrospray ionization (nano-DESI)[49], two capillary probes are used to deliver solvent to/from a sample surface for extraction/ionization. SC-MSI of tissues at a sub-10 μm resolution[50], buccal cells[51] (pixel size of 150 μm), and IMR-90 cells[52] (pixel size of 50 μm) has been reported through the use of sharp probes and feedback control techniques.

We previously developed tapping-mode scanning probe electrospray ionization (t-SPESI). In t-SPESI, a vertically oscillating capillary probe is used to perform rapid local extraction and ionization of tissue components[53,54]. A feedback control technique for probe oscillation amplitude was developed to stabilize probe scanning and simultaneous acquisition of ion and topographic images when performing MSI on uneven tissue sections[55,56]. In this paper, we introduce a novel t-SPESI measurement system for SC-MSI and visualization of the lipid distribution in a single HeLa cell.

To achieve SC-MSI via t-SPESI, elemental technology to reduce the size of the extraction area to a few micrometers (subcellular scale) and measure ions with high sensitivity via a mass spectrometer must be developed. In order to realize this, we set the following three objectives in technological development:

### Development of a high-precision multimodal imaging system
Cells cultured on a glass substrate show a random distribution unless a patterned substrate is used. In addition, the morphology of cells is not flat and involves micrometer-to-submicrometer irregularities[57,58]. Therefore, a system to determine the location of cells and perform extraction–ionization with microscale precision is required. We aimed to develop a t-SPESI unit that can be used in conjunction with an inverted fluorescence microscope to obtain optical, ionic, and topographic images of cells cultured on glass substrates.

### Stabilization of the solvent flow to form micro liquid bridges
The solvent volume required to form a hemisphere with a radius of 1 μm on a solid surface was estimated to be approximately 2 fL. To form liquid bridges with diameters of several micrometers, we developed a solution to prevent clogging of the capillary probe and introduced a nanoflow pump to perform solvent pumping at low flow rates.

### Modification of the ion transfer tube to realize high-sensitivity measurement of ions
As the extraction area decreases, the number of produced ions also decreases. Therefore, a solution to transfer ions to the mass spectrometer with as little loss as possible is required. We investigated the structure of the ion transfer tube used to introduce ions into a mass spectrometer and measure ions with high sensitivity.

## Results and discussion
### Development of a t-SPESI system for high-precision multimodal imaging
A block diagram of the newly developed measurement system is shown in Fig. 1a. Cells cultured on glass substrates were mounted on a sample stage of an inverted fluorescence microscope. The cells were stained with fluorescent dyes, chemically fixed, and dried. A capillary probe was mounted on a holder with the piezoelectric actuator of the t-SPESI unit and vertically oscillated by applying a sine-wave voltage signal to the actuator. A high voltage was also applied to the metal union connected to the probe to charge the solvent. The solvent was supplied from a nanoflow pump to the probe. When the probe tip contacted the sample, a liquid bridge formed between them and the sample components were extracted. Next, when the probe was separated from the sample, the extracted solution was lifted up and transformed into charged droplets via electrospray. The charged droplets dried while passing through the heated ion transfer tube, and the molecules in the droplets were converted to gas-phase ions. These ions were introduced into the mass spectrometer through an orifice, and mass spectra were obtained. Figure 1b shows a photograph of the arrangement of the t-SPESI unit, sample, and ion transfer tube.

The surface of a single cell is uneven. Therefore, if the probe is scanned over the cell with a fixed distance between the probe tip and the sample surface, the distance between the probe and cell surface changes depending on the sample shape, resulting in variations in the oscillation amplitude. This would change the contact time between the probe and the sample,

which might cause changes in the ion signal intensity. Therefore, a method to maintain a constant oscillation amplitude during probe scanning is required. To measure the oscillation amplitude of the probe, a laser beam was irradiated on the side of the probe, the transmitted light was sensed by a two-segment photodiode, and the vertical displacement of the probe shadow was detected by measuring the voltage difference between the segments. The displacement signal was input to a lock-in amplifier, and the oscillation amplitude was measured.

In the new t-SPESI unit, two mirrors were used to bend the laser beam, and the laser source and photodiode unit positions were changed. This allowed the t-SPESI unit to be placed above the large sample stage of the inverted fluorescence microscope without interference. A rendered image of a 3D model of the t-SPESI unit and a photograph of the actual device are shown in Supplementary Fig. 1. The t-SPESI unit was connected to the piezo stage, and the vertical position of the probe was controlled. The relationship between the output signal from the feedback control system and the sample height is shown in Supplementary Fig. 2. The probe position could be dynamically adjusted to account for changes in the sample height in the range of 10 μm.

The presented measurement system, in which an inverted fluorescence microscope and a t-SPESI unit are combined, allows both the sample and the liquid bridge at the probe tip to be observed from the back side of the sample. Figure 1c shows a bright-field image acquired during a measurement. HeLa cells cultured on glass substrates were observed on the right side of the image. As shown by the arrows, a liquid bridge with a darker contrast than the glass substrate was observed. To evaluate the spreading of the liquid bridges formed on the glass substrate, bright-field images were obtained and analyzed. The average diameter of the liquid bridges was 4.0 μm, with a standard deviation of 0.87 μm when MSI was performed on HeLa cells. (Supplementary Fig. 3 and Table 1). This result shows that the area of the liquid bridge varied during the measurements. The factors causing this

variation are possibly the fluctuations in the solvent flow and differences in the local wettability of the glass surface. The ion images presented here were acquired using the oversampling method[59,60], because the size of the liquid bridge was larger than the pixel size.

To realize SC-MSI using t-SPESI, elemental technologies to reduce the size of the liquid bridge are necessary. We used the measurement system combined with an inverted microscope to directly observe the liquid bridge from the backside while performing the MSI. We confirmed that the solvent did not spread significantly over the glass substrate or cells during probe scanning. To the best of our knowledge, there have been no reports of techniques for observing the extraction area from the backside of the sample using other direct extraction ionization methods, such as DESI and nano-DESI, during MSI. In addition, syringe pumps are often used for solvent flow, but the issue with syringe pumps is that it is difficult to monitor the pressure and flow rate of the solvent. Therefore, we used a nanoflow pump to confirm that the solvent flow rate and pressure were maintained while performing MSI. We considered that microscale extraction could be achieved by combining both direct observation of the liquid bridge using a microscope and monitoring the status of the nanoflow pump.

## Probe design for stabilization of the solvent flow to form micro liquid bridges

To realize SC-MSI with t-SPESI, a technique to supply a solvent from a capillary probe to a micrometer-sized region of a sample is essential. Because the diameter of the liquid bridge at the probe tip depends on the tip aperture, the tip aperture of the probe must be reduced to a few micrometers. The aperture can be reduced using a laser puller, but the flow path inside the probe is easily clogged by impurities in the solvent. We solved this issue by filling the capillary probe with amino-group-modified silica particles (N-particles) with an average diameter of 3 μm. Figure 2a shows a typical particle-filled capillary probe. Approximately 3.6 mm from the tip, the flow

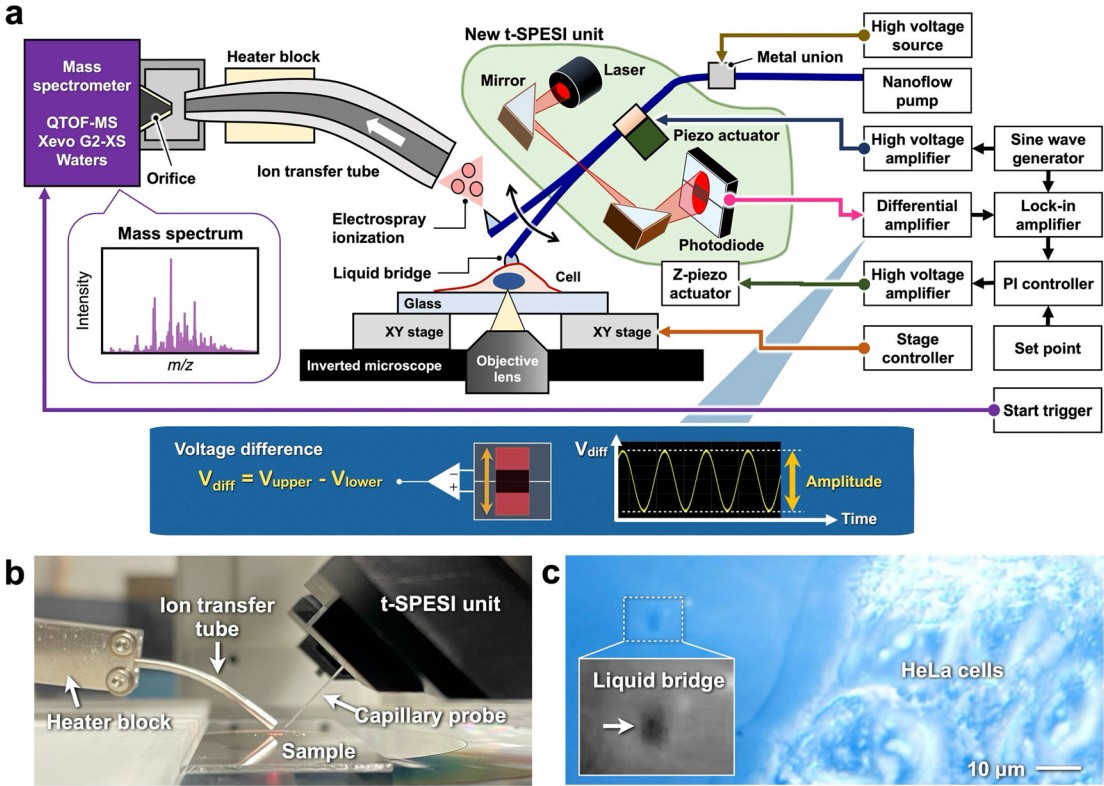

**Fig. 1 | Development of the t-SPESI system with an inverted microscope for single-cell MSI. a** Schematic illustration of the proposed t-SPESI system. **b** Image of the t-SPESI unit and ion transfer tube. **c** Optical microscope image of a liquid bridge and HeLa cells on a glass substrate. A magnified image of the liquid bridge is shown inside the figure.

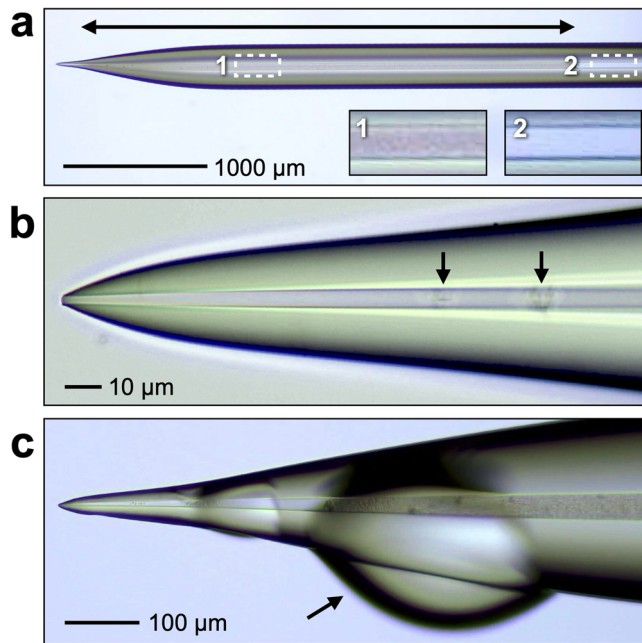

**Fig. 2 | Observation of the capillary probe packed with silica particles. a** Low-magnification image of the capillary probe. The arrowed line indicates the region in which the silica particles were packed inside the capillary probe. **b** High-magnification image of the capillary probe. The arrows indicate the positions of the silica particles inside the capillary probe. **c** View of solvent flow through the probe. When methanol was flowed inside the probe, it flowed out from the probe end and accumulated on the side of the probe (arrows in the figure).

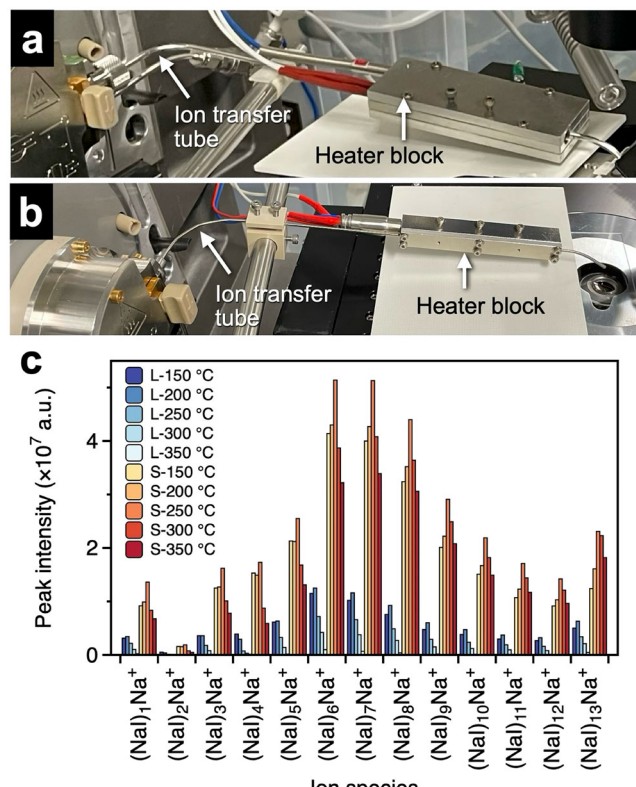

**Fig. 3 | Photograph of the ion-transfer tubes. a** Tube with an O.D. of 1/8 inches and an I.D. of 2.18 mm, and **b** tube with an O.D. of 1.61 mm, and an I.D. of 1.25 mm. The end of the ion transfer tube was connected to the orifice of the mass spectrometer. A heater block was attached to the tube. **c** Comparison of NaI cluster ion signal intensities. L and S refer to the ion transfer tubes in (**a**) and (**b**), respectively, and the numbers indicate the temperature of the tube.

path was filled with N-particles (region indicated by the arrowed line in the figure). The area filled with particles (area 1 in the figure) showed a change in contrast compared with the area without particles (area 2 in the figure). An enlarged view of the tip is shown in Fig. 2b. The aperture of the probe tip was approximately 2 μm, and a small number of particles were trapped in the flow path approximately 100 μm from the tip (region indicated by arrows in Fig. 2b). When MeOH was injected inside the probe, MeOH flowed out from the probe tip (Fig. 2c). The arrow in Fig. 2c points to MeOH that flowed out of the probe tip and accumulated on the side of the probe.

Silica capillaries packed with octadecylsilyl-group-modified silica particles (O-particles) can be used for proteomics separation[61]. We also attempted to use probes packed with O-particles, but the particles moved in the flow path upon application of a positive voltage to the solvent, and the solvent flow stopped over time. The surface of the O-particles contained some silanol groups, and the internal surface of the probe also contained silanol groups. The introduction of the solvent inside the probe was thought to have induced negative charges on the silanol groups, resulting in electrostatic repulsion between the O-particles and the flow path, causing the O-particles to detach from the inner surface and move toward the metal union biased with a positive voltage (Supplementary Fig. 4). In contrast, when the N-particles were filled, the particles were fixed in the flow path without being affected by the application of a high voltage. Furthermore, the position of the N-particles did not significantly change even when the probe oscillated in the vertical direction (Supplementary Fig. 4). This result may be due to the formation of hydrogen bonds between the amino groups of the N-particles and the silanol groups in the channel, which suppressed particle migration even under high-voltage application and probe oscillation.

With respect to stabilization of the solvent flow, monitoring of the actual state of the solvent flow was previously difficult because a syringe pump was used[56]. Using a nanoflow pump, the flow rate (1 nL min⁻¹) during the MSI experiment was monitored. Given the oscillation frequency of the

probe (656–657 Hz), the volume of solvent consumed in a single extraction–ionization cycle was estimated to be 25 fL. Considering that the diameter of the liquid bridges was 4 μm, approximately 67% of the solvent supplied during a single probe oscillation cycle (1.5 ms in this case) was used to form the liquid bridges. In t-SPESI, an oscillating probe is used to chop the solvent flow supplied at a constant rate, which allows the formation of minute liquid bridges for extraction.

### Investigation of ion transfer tube to realize high-sensitivity measurement of ions

When SC-MSI is conducted with t-SPESI, the volume of solvent consumed per extraction–ionization event is on the order of several tens of femtoliters, and the number of generated ions is small. To introduce ions into a mass spectrometer with high efficiency via a differential pumping system, we investigated the effect of the inner diameter of the transfer tube on the ion signal intensities. Tubes with inner diameters of 2.18 mm (type L) and 1.30 mm (type S) were bent into an L shape and connected to the orifice (Fig. 3a, b). To facilitate desolvation, a heater block was fixed near the tube inlet.

ESI of a NaI solution was conducted to compare the signal intensities of the cluster ions (Fig. 3c). Although NaI cluster ions were confirmed for both transfer tubes, the signal intensity increased when the S-type tube was used. The ion signal intensity varied with the temperature of the ion transfer tube, with the highest signal intensity obtained when the tube was heated to 250 °C. The mass spectra obtained at different temperatures and with different tubes are shown in Supplementary Fig. 5.

The results suggested that the reduction of the cross-sectional area of the tube flow path to 36% improved the convergence of ions passing through the transfer tube, resulting in an efficient introduction of ions into the mass

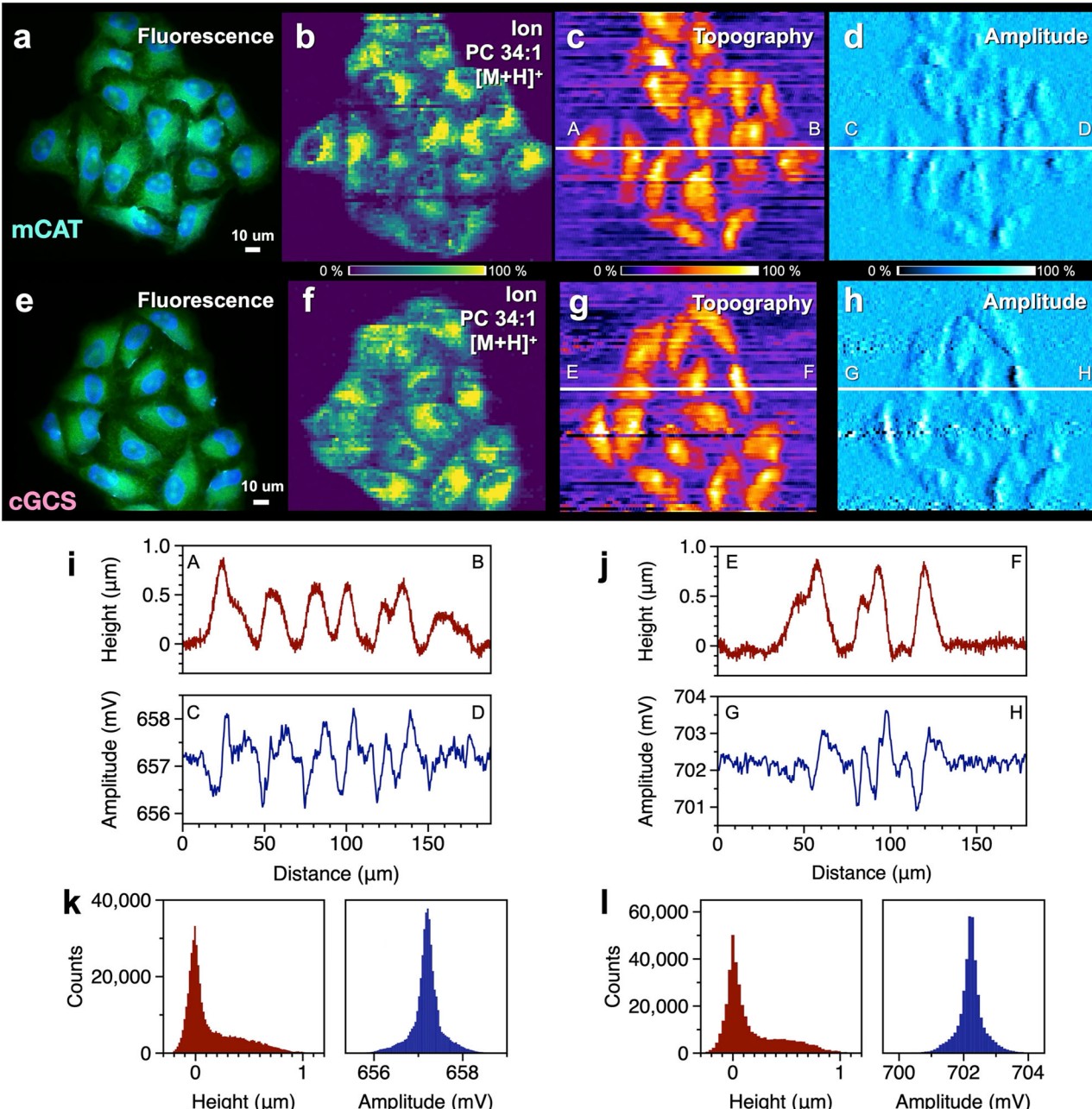

**Fig. 4 | Results of multimodal imaging of HeLa cells using t-SPESI.** Fluorescence (**a**, **e**), ion (**b**, **f**), topographic (**c**, **g**), and amplitude (**d**, **h**) images of mCAT and cGCS cells. **i**, **j** Section profiles of topographic and amplitude images of mCAT and cGCS cells, respectively. The positions of the cross sections are indicated in the topographic and amplitude images. **k**, **l** Histograms of the signal intensities of the topographic and amplitude images of mCAT and cGCS cells, respectively.

spectrometer. The increase in the signal intensity with increasing temperature up to 250 °C can be mainly attributed to the enhanced desolvation of the charged droplets. Inside the ion transfer tube, the detection sensitivity of ions is likely to have been improved by effectively removing large droplets and enhancing desolvation by collisions between the charged droplets and the inner wall of the tube. The decrease in the ion signal intensity with increasing temperature above 250 °C can be attributed to the decrease in the effective gas velocity inside the transfer tube owing to thermal choking caused by heating and to the decrease in the number of ions owing to collisions with the inner wall of the transfer tube[62]. Ion transfer through an L-shaped tube is a complex process. To determine the optimal shape of the ion transport tube, computational fluid dynamic simulations[63] to investigate the effects of charged droplets, ions, atmospheric gas flow, and space charge must be conducted in the future.

## Multimodal imaging of HeLa cells

The following section describes the SC-MSI results of HeLa cells obtained using the new t-SPESI system. In this study, we examined HeLa mCAT cells and HeLa IP-cGCS (cGCS) cells overexpressing glucosylceramide synthase to distinguish between cell types and compare the distribution of intracellular lipids. A mixture of DMF and MeOH was used to measure lipid ions[56]. We previously reported that this solvent mixture can suppress the disappearance of biological tissue, and that it is possible to perform hematoxylin and eosin staining of the tissue after MSI and compare the cell distribution and lipid distribution of the same tissue[64].

A fluorescence image of HeLa mCAT cells is shown in Fig. 4a. Multiple cells were present on the glass substrate without any overlap, and the cytoplasm and nucleus were identified via fluorescent molecules. An ion image of the same cells is shown in Fig. 4b. The pixel size was set at 2 μm. The

distribution of the signal intensity of PC 34:1 $[M + H]^+$, which is abundant in the cells, is shown. In the topographic image (Fig. 4c), the height increased in regions where cells were present, particularly in the area corresponding to the cell nucleus. In the amplitude image (Fig. 4d), which shows the change in the probe oscillation amplitude, different contrasts were observed on the left and right sides of a cell due to a delay in feedback control. This image shows that the amplitude changed at positions where the height of a cell significantly changed while the probe was scanned over the cell. Notably, such contrast in amplitude images is commonly observable in tapping mode atomic force microscopy (AFM)[57]. Similar results were obtained for cGCS cells (Fig. 4e–h).

Cross-sectional profiles of the topographic and amplitude images of each cell are shown in Fig. 4i, j. The height of the cell from the substrate was approximately 500 nm to 1 μm, and the amplitude increased or decreased at the cell edges as the probe was scanned over the cell. Histograms of the sample heights and amplitudes for all the pixels in the topographic and amplitude images are shown in Fig. 4k, l. In the sample height histograms, the glass substrate region was shown as a peak centered at a relative height of 0 μm, whereas the height of the cell region presented a broad distribution in the range of up to 1 μm. In contrast, the amplitude histograms showed a sharp single peak, which corresponded to the set-point value of the amplitude for the feedback control. The difference between the maximum and minimum amplitude values (approximately 4 mV) can be attributed to a delay in the feedback control. However, the variation with respect to the central value was small (less than 1%), suggesting that the feedback control worked properly. If the feedback control had a negative effect, the signal intensity would be expected to vary depending on the relative position of the nucleus and cytoplasm; however, similar ion images were obtained for each cell. For example, in Fig. 4b, f, the signal intensities of the cell nuclei were lower and the intensities of the cytoplasm were higher. These results successfully demonstrated multimodal (fluorescence, ion, and topographic) imaging of HeLa cells with a subcellular pixel size (2 μm) via t-SPESI.

The oscillation amplitude changes depending on the distance between the capillary probe and sample[65]. In this study, SC-MSI data were obtained using the same probe. The correlation between the probe oscillation conditions (amplitude and frequency), size of the liquid bridge, and ionization efficiency should be investigated in future studies.

## Distinguishing cells by lipid information

We examined whether the lipid information acquired via SC-MSI was usable in distinguishing mCAT cells from cGCS cells (Fig. 5a). The ion peaks obtained from the measurements were matched with the LIPID MAPS database, and 166 ion peaks tentatively assigned to a single lipid ion were selected. Because the ion signal intensity in this measurement system is currently insufficient for tandem mass spectrometry (MS/MS), lipidomic analysis via SFC-MS/MS was performed to evaluate the lipids in HeLa cells.

Next, ROIs were set for 45 individual mCAT and cGCS cells from the nine MSI datasets for different cell clusters. To compare the differences in the mass spectra of different cell types, the mass spectra of all the ROIs were averaged for each cell type (Supplementary Fig. 6). Ion peaks derived from lipids were observed in the $m/z$ range of 650–900. No obvious differences were observed in the overall mass spectrum patterns; thus, the signal intensities of the individual ion peaks were compared. The signal intensities of the lipid ions from the average mass spectra of the ROIs were used to conduct Welch's $t$-test, and 29 lipid ions with $p$ values less than 0.005 were selected (Supplementary Table 6) to prepare a volcano plot (Fig. 5b). Principal component analysis was applied to the intensity data of these lipid ions to obtain a score plot. Most cells were classified into two groups with 95% confidence ellipses (Fig. 5c).

Box plots to compare the ion intensities of 90 cells for the six lipid ions are shown in Fig. 5d–i. The selected lipids are shown in a volcano plot (triangles in Fig. 5b). The box plots of all 29 lipid ions in the volcano plot are shown in Supplementary Fig. 7. The ions with $m/z$ values of 758.569, 787.669, and 749.560 were assigned to PC 34:2 $[M + H]^+$, SM d18:1/22:0 $[M + H]^+$, and SM d18:1/18:2 $[M + Na]^+$, respectively. The signal

intensities of these ions were significantly greater in mCAT cells than in cGCS cells. In contrast, ions with $m/z$ values of 848.556, 741.532, and 784.667 were assigned to PC 38:4 $[M + K]^+$, SM d18:1/16:0 $[M + H]^+$, and HexCer d18:1/22:0 $[M + H]^+$, respectively. The signal intensities of these ions were significantly greater in cGCS cells than in mCAT cells.

Next, the results of the comparison of the ion images of the six lipids with large differences in signal intensity, shown in box plots, are presented. Figure 5j–l shows lipid ion images in which the signal intensity of mCAT-cells is greater than that of cGCS cells. In contrast, Fig. 5m–o shows lipid ion images in which the signal intensity of cGCS cells is greater than that of mCAT cells. The color scales for each ion image were unified to compare the relative differences in the signal intensities of the different cell types. Different signal intensity distributions within the cells were clearly observed in each ion image. The other ion images of HeLa cells are shown in Supplementary Fig. 8.

Figure 5p–w shows magnified fluorescence and ion images of mCAT and cGCS cells, respectively. In the ion images of PC 34:2 $[M + H]^+$ and PC 38:4 $[M + K]^+$, the signal intensity of the cell nucleus was low, and the signal intensity of the lateral local region of the nucleus increased (Fig. 5q, u). The phospholipid composition of cell membranes reportedly differs depending on the organelle. The endoplasmic reticulum and Golgi apparatus are near the nucleus, and PC is the most abundant component, followed by PE[66]. In another fluorescence image of HeLa cells used in this study, the Golgi apparatus was observed to be located on the side of the nucleus (Supplementary Fig. 9). The localization of PC in ion images is thought to derive from intracellular organelles. The presented results show that the amount of lipids in the nucleus is low and qualitatively consistent with the results of other imaging techniques. Phospholipids in the nucleus are mainly present in the nuclear membrane, and it has been reported that in rat liver cells, the main components inside the nucleus are proteins and DNA[67]. The results of single-cell MSI using MALDI have shown that the signal intensity of phospholipids decreases in the nucleus of Vero B cells[37]. In addition, a decrease in Raman signals derived from lipids in the cell nucleus has been reported in observations of HeLa cells using stimulated Raman scattering microscopy[68].

In contrast, in the SM d18:1/22:0 $[M + H]^+$, SM d18:1/18:2 $[M + K]^+$, and SM d18:1/16:0 $[M + K]^+$ images, there was no clear localization similar to that of PC, and the signal was distributed over a wider area than that of PC (Fig. 5r, s, v). SMs are produced in the Golgi apparatus and are transported to the cell membrane via vesicular transport. SMs and their metabolites, which are located outside the cell membrane, are important active molecules involved in many signal transduction processes, such as cell proliferation, differentiation, aging, and apoptosis[69,70]. The ion images of SMs, which have a different distribution pattern than that of PC, are thought to indicate the SMs in the apical membrane distributed throughout the cell. The ion images of HexCer revealed an increased signal intensity in cGCS cells, which had high expression of glycolipid synthase (Fig. 5w). However, the signal intensity was generally low, and no clear localization was observed within the cells. MSI of HeLa cells via t-SPESI not only revealed that the signal intensity significantly differed between cell types but also revealed that the range of the signal intensity greatly varied between cells (Fig. 5d–i). One possible reason for this is the change in the lipid content that occurs during the cell cycle. In future research, we will consider performing MSI of cell cycle-dependent metabolites and lipids via fluorescent labeling[71] to visualize the cell cycle.

The present results show that t-SPESI can be used to obtain molecular information within cells. However, notably, realizing quantitatively accurate SC-MSI comparable to lipidomic analysis via chromatographic separation techniques is still challenging. Similar to the ionization methods used in other MSI techniques, the sample components were simultaneously extracted and ionized in t-SPESI. Thus, ion suppression is thought to occur because of multiple factors, such as the extraction efficiency of the solvent, owing to the physicochemical properties of the molecules and the cation content, which are thought to affect the signal intensity of the ions. We confirmed that the signal intensity ratios of lipid ions between the cell types

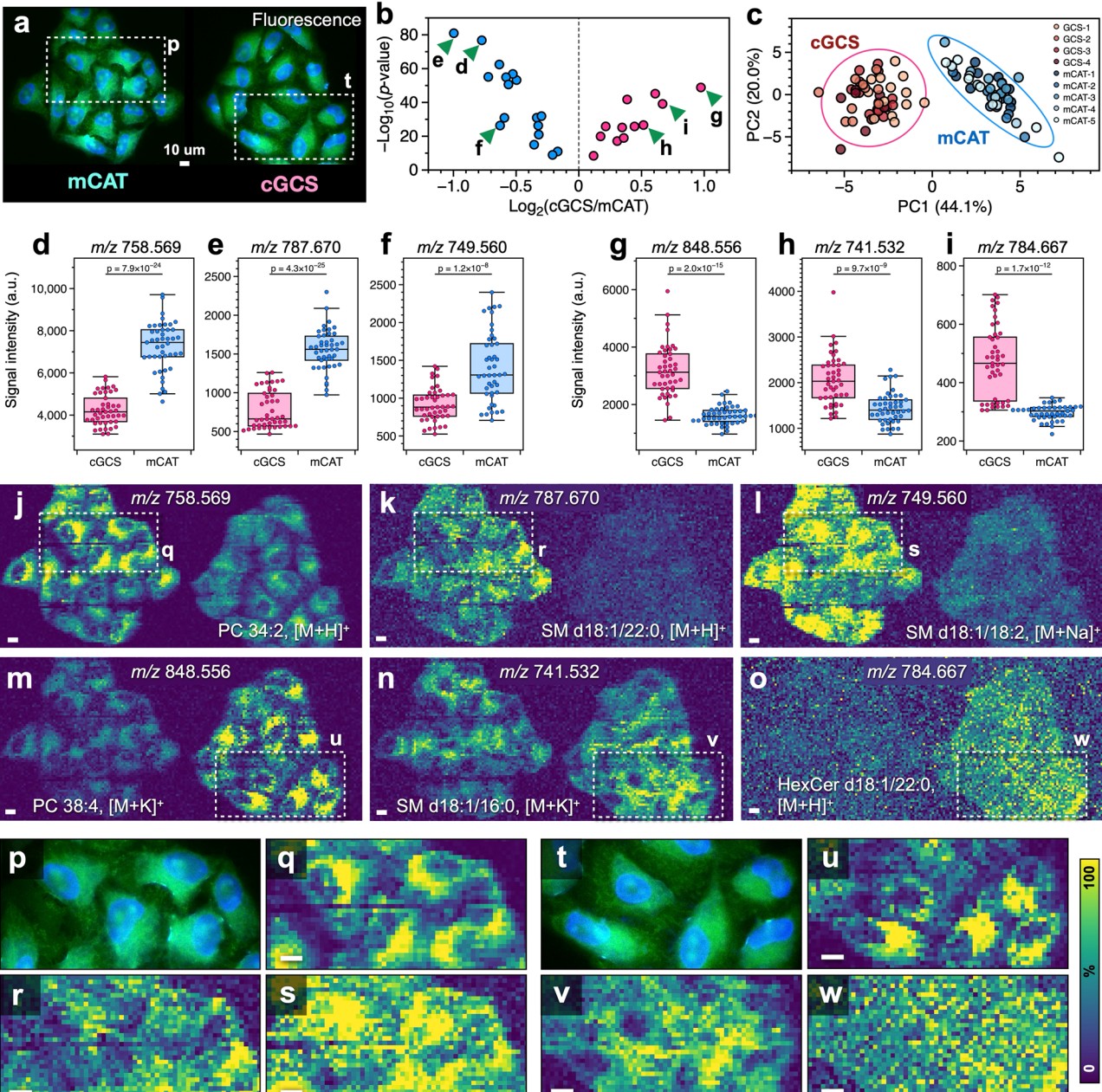

**Fig. 5 | Results of distinguishing cell types based on the signal intensities of lipid ions. a** Comparison of fluorescence images of mCAT and cGCS cells. **b** Volcano plot showing differences in the signal intensities of lipid ions between cell types. **c** Score plot obtained via principal component analysis. **d–i** Bar plot showing the differences in the signal intensities of lipid ions in single cells. **j–l** Comparison of ion images with high signal intensities in mCAT cells. **m–o** Comparison of ion images with high signal intensities in mCAT cells. In each ion image, the left side shows mCAT cells, and the right side shows cGCS cells. The *m/z* values of the ions and the assigned lipid ions are shown in (**j–o**). **p–s** Enlarged fluorescence and ion image showing a high signal intensity in mCAT cells. **t–w** Enlarged fluorescence and ion image showing a high signal intensity in cGCS cells. The enlarged regions are indicated by dashed lines in (**j–o**). Scale bar: 10 μm.

measured via t-SPESI MSI and the lipid content ratios between the cell types quantified via SFC-MS/MS did not necessarily match (Supplementary Fig. 10). One of the most important issues in SC-MSI is ensuring quantitative analysis[20,72], and the addition of an IS to the solvent[73] is expected to be a solution. To ensure accurate signal normalization, it is necessary to use a sample that is chemically and morphologically well-defined, rather than a complex sample with multiple components, such as cells or biological tissue. Furthermore, it is necessary to consider a data analysis method that comprehensively evaluates the topography, amplitude image, and ion image using a solvent to which an internal standard has been added.

Under the probe oscillation conditions used in this study, it was confirmed that the trace of probe scanning remained after measurement

(Supplementary Fig. 11). Thus, we consider that the probe penetrated the interior of the sample and performed extraction and ionization. Optimization of the oscillation frequencies is important for improving the spatial resolution and increasing the ion signal intensity. To improve the spatial resolution, it is necessary to use a capillary probe with a finer tip and further reduce the size of the liquid bridge. However, as the solvent volume decreased, the number of extracted molecules decreased. One conceivable method to overcome this problem is to increase the oscillation frequencies for the extraction-ionization event, thereby preventing a decrease in the ion detection sensitivity. In t-SPESI, the resonance frequency of the capillary probe is proportional to the reciprocal of the square of the probe length[55]. Therefore, the resonance frequency can be increased by reducing the probe length.

To realize SC-MSI, we developed an advanced t-SPESI unit that can be used in combination with fluorescence microscopy. We demonstrated multimodal imaging of the same cell cluster, including fluorescence, ion, and topographic imaging. Two types of HeLa cells could be distinguished based on the signal intensities of multiple lipid ions, and the characteristic intracellular lipid distributions were visualized. Compared with conventional MSI, SC-MSI is expected to significantly improve the quality of spatial omics information of biological tissues and complement imaging using fluorescence microscopy and cell analysis using flow cytometry. The evolution of SC-MSI, which can be used to measure the multidimensional molecular information of individual cells that make up diseased tissue and/or have been subjected to external stress, is expected to contribute to the derivation of new knowledge for biomarker discovery, elucidation of pathogenesis, and heterogeneity in metabolic responses. Techniques for arranging cells in a regular pattern on a patterned substrate have been used to obtain physical and biochemical information to understand cell behavior, such as cell morphology, differentiation, and apoptosis[74]. By combining these technologies with SC-MSI, cell metabolites and responses to drug administration can be profiled at the molecular level under well-defined conditions.

## Methods
### Preparation of HeLa cells
HeLa mCAT cells and Chinese hamster glucosylceramide synthase-overexpressing HeLa cells (HeLa IP-cGCS)[75] were gifted by Dr. Toshiyuki Yamaji. The cells used in this study were not classified as commonly misidentified lines because they were derived from HeLa cells. HeLa mCAT and HeLa IP-cGCS cells were cultured with low-glucose Dulbecco's modified Eagle's medium (DMEM) containing 10% fetal bovine serum (FBS) and 1× penicillin−streptomycin. The cells were cultured in 6-well plates (w3810-006N, AGC TECHNO GLASS, Japan.) with cover glass (18 mm × 18 mm, No. 1, Matsunami, Japan) at a concentration of $0.4 \sim 1.0 \times 10^4$ cells mL$^{-1}$ for approximately 24 h. The cell culture time was adjusted to use HeLa cells in a preconfluent state. All cell lines tested negative for mycoplasma contamination by PCR. It was confirmed that there was no overlap between the adjacent HeLa cells by observing the cells using a fluorescence microscope. The low-glucose DMEM with L-glutamine and phenol red (041-29775) and penicillin−streptomycin solution (×100) (168-23191) were purchased from FUJIFILM Wako Pure Chemical Corporation, Japan. The FBS (35-076-CV) was purchased from Corning, USA.

### Pretreatment of HeLa cells for MSI
Hoechst 33342 (1 mg mL$^{-1}$, H3570, Invitrogen, USA) and fluorescein diacetate (FDA, 25 mg mL$^{-1}$, F7378-5G, Sigma-Aldrich, USA) were dissolved in phosphate-buffered saline (PBS, 164-28713, FUJIFILM Wako Pure Chemical Corporation) to final concentrations of 1 µg mL$^{-1}$ and 50 µg mL$^{-1}$, respectively. As a cell fixation solution, 2.5% glutaraldehyde in PBS solution (072-02262, FUJIFILM Wako Pure Chemical Corporation) was diluted 10-fold in PBS and used[76]. The medium in the well plates was removed via an aspirator (Nichimate ASPIRATOR IIa, Nichiryo, Japan), 2 mL of PBS was added, and the cover glass on which the HeLa cells were cultured was picked up. One milliliter of PBS was applied three times in succession to the HeLa cells, and the excess solution was removed via a KimWipe. A premixed dye solution (1 mL) was applied and incubated under light-shielded conditions for 5 min. Next, 1 mL of PBS was applied three times, the excess solution was removed, and 1 mL of the fixation solution was applied and incubated for 5 min. Finally, 1 mL of ammonium acetate was applied twice, and the excess solution was removed via an air blower and allowed to dry. To visualize the nucleus and Golgi apparatus in HeLa cells, the nucleus was stained with Hoechst 33342, and red fluorescent protein for binding to N-acetylgalactosaminyltransferase in the Golgi apparatus was expressed.

### Preparation of capillary probes
The polyimide film covering a fused silica capillary (TSP030375, outer diameter (O.D.) of 360 µm, inner diameter (I.D.) of 30 µm, Molex, USA) was seared with a gas burner and removed via an ethanol-soaked Kim-Wipe. A capillary probe with a tip aperture of approximately 2 µm was fabricated via a laser puller (P-2000, Sutter, USA). The parameters set for P-2000 are listed in Supplementary Table 2. After processing, the tip of the capillary probe was observed via a upright microscope (Eclipse Si, Nikon, Japan).

### Packing of silica beads into capillary probes
Silica particles modified with amino groups (ReproSil-Pur 1000 NH2, average diameter of 3 µm, Dr. Maisch GmbH, Germany) or octadecyl groups (ReproSil-Pur 120 C18-AQ, average diameter of 3 µm, Dr. Maisch GmbH) were used. Approximately 300 mg of silica particles and 1 mL of a solvent mixture (N,N-dimethylformamide (DMF)/methanol (MeOH) 1/1 v/v) were mixed in a glass vial. DMF (high-performance liquid chromatography (HPLC) grade, 13024-71, Nacalai Tesque, Japan) and MeOH (HPLC grade, 21929-81, Nacalai Tesque) were used. The suspension was centrifuged (11,000 rpm for 60 s), the supernatant was removed, and the mixture was redispersed in a mixed solvent. This procedure was repeated ten times, and the sample was finally redispersed in 300 µL of the mixed solvent. A capillary probe was cut approximately 5 mm from the end, and the end was immersed in the silica particle dispersion and left for 45 s. A PEEK male nut (JR-55020-5, Shimadzu GLC, Japan) and an FEP sleeve (F-376, IDEX, Japan) were used to connect the capillary probe to a PEEK union (6010-48740, GL Science, Japan). A PEEK tube (O.D. of 1/16 inches, I.D. of 75 µm, JR-T-5803, Shimadzu GLC), a PEEK luer adapter (P-659, IDEX), and a syringe (1725TLL, Hamilton, USA) were sequentially connected to the other port of the union via a PEEK male nut. MeOH was manually flowed from the syringe while the probe tip was observed under a microscope to confirm that the silica capillary near the probe tip was filled with silica particles and that MeOH flowed out.

### Calibration of the piezo actuator used for feedback control
A step master (516-198, Mitsutoyo, Japan) was mounted on the sample stage, and probe scanning was conducted over two regions of different heights. The output signal of the PID controller was recorded during probe scanning. The relationship between the height difference and PID output signal was linearly fitted to obtain the calibration function.

### Examination of the ion transfer tubes
Two types of stainless steel ion transfer tubes with different inner diameters (type L: O.D. of 1/8 inch, I.D. of 2.18 mm; type S: O.D. of 1.61 mm, I.D. of 1.25 mm, length 27 cm, material: SUS 304, Kuroiwa stainless steel industry Inc., Japan) were used. The transfer tube was fixed in front of the sampling cone (orifice) of a quadrupole time-of-flight mass spectrometer (Xevo G2-XS qTOF, Waters, USA) using a custom-made adapter. The opening of the transfer tube was placed above the sample stage, and the position of the ion transfer tube was adjusted using homemade parts that held the tube on an XYZ manual stage. NaI powder (99.5%, Wako Special Grade, FUJIFILM Wako Pure Chemical Corporation) was mixed with water/isopropanol (50/50 v/v) at a concentration of 2 µg µL$^{-1}$. A syringe pump (Legato 185, KD Scientific, USA) was used to pump the NaI solution into a fused silica capillary (O.D. of 363 µm, I.D. of 25 µm, TSP020375, Molex) at a flow rate of 300 nL min$^{-1}$. The capillary end was placed in front of the transfer tube, and 5 kV was applied to the metal needle of the syringe to perform measurements in positive ion mode (sensitivity mode). The temperature of the heater block fixed to the transfer tube was controlled from 150 °C to 350 °C by a temperature controller (MTCS, Misumi, Japan). The mass spectra were measured for 0.5 min for each temperature condition. The intensities of the NaI cluster ion peaks in the mass spectra were analyzed via MassLynx (Waters). The m/z values and ion types of the cluster ions considered in this study are listed in Supplementary Table 3. The boiling point of DMF (153 °C) at 1 atm is higher than that of water (100 °C). For the SC-MSI of HeLa cells, the temperature was set to 300 °C to enhance desolvation. This value was 50 °C higher than the temperature that showed the maximum value in the measurement of the NaI solution.

## t-SPESI measurement system

The new measurement system consisted of a t-SPESI unit, a microscope unit, and a control unit. In the t-SPESI unit, an enclosure containing a laser source, a piezo actuator for probe excitation, and a laser light detection device was fixed to the piezo actuator (B22-083, THK precision, Japan). The XYZ step motor stage connected to the t-SPESI unit was used to align the probe and sample. A commercial inverted fluorescence microscope (TE2000-U, Nikon, Japan) with a 40x dry objective lens and a high-precision linear stage (BIOS-206T, Opto Sigma, Japan) was used as the microscope unit. The optical images were obtained with a digital camera (DS-Fi3, Nikon, Japan) which was mounted on the fluorescence microscope. The camera was controlled by a software (NIS Elements, Nikon, Japan). The stage position was controlled at a resolution of 0.1 µm. The probe oscillation and scanning were controlled via a custom-made program coded in LabVIEW (NI, USA). The instruments used are listed in Supplementary Table 4.

## SC-MSI

A glass substrate with HeLa cells was mounted on a sample holder (92001, Chroma Technology Corp., USA). The holder was then fixed to the sample stage of the fluorescence microscope. An equivalent mixture of DMF and MeOH by volume, with 0.1% formic acid, was used as the solvent. For data acquisition, the probe was scanned at a constant speed from the upper-left position of the measurement area (corresponding to the ion image) in the right direction. The scanning speed of the probe was $6.7 \ \mu m \ s^{-1}$. At the beginning of each single line scan, a trigger signal was input into the mass spectrometer to start the measurement. When a single line scan was completed, the probe returned to the starting position of the scan and then moved down 2 µm for the following scan. Mass spectra were acquired in positive ion mode over the $m/z$ range of 100–1200 with a 300 ms accumulation time. The mass resolving power of the mass spectrometer was >30,000 FWHM in sensitivity mode. The pixel size is defined by the distance the probe scans over a given time in the X-direction and the distance between the scanning lines in the Y-direction. Here, the X-direction corresponds to the scanning direction of the probe (fast scan axis), and the Y-direction corresponds to the direction orthogonal to the scanning direction of the probe (slow scan axis). In this study, the pixel size of the ion images was 2 µm in both the horizontal and vertical directions. On average, the signal intensity of a single pixel in an ion image corresponds to the sum of the ion signals measured in 197 extraction–ionization events.

The solvent flow was controlled by a nanoflow pump (LC-20AD nano, Shimadzu). The flow rate and pressure of the solvent were $1 \ nL \ min^{-1}$ and 0.7–0.9 MPa, respectively. The voltage applied to the solvent through the stainless steel union was adjusted to approximately 2.6 kV. The set point of the oscillation amplitude was 90% of the free oscillation amplitude before the probe approached the sample. The ion transfer tube was heated to 300 °C. The temperature of the mass spectrometer orifice was set at 150 °C.

Before MSI, mass calibration was performed by ionizing a polyalanine film. The film was prepared by dissolving 4.5 mg of polyalanine (P-9003, Sigma−Aldrich, USA) in 1 mL of MeOH/H$_2$O solvent (95/5 v/v), and the solution was dropped onto a glass slide (S1126, Matsunami), which was precleaned with pure water and methanol sequentially for 10 min using an ultrasonic cleaner (ST01D, Sonic Tech) and dried on a hot plate at 70 °C (CHP-170DF, AS ONE, Japan). To reduce the background ion signal caused by components in the laboratory atmosphere, clean air was supplied to the area near the inlet through a Teflon tube connected to an active background ion reduction device (ABIRD, ESI source solutions, USA).

We obtained a snapshot of the bright-field image to evaluate the spreading of liquid bridges. ImageJ was used to measure the area by enclosing the area with an ellipse. The diameter was calculated by assuming that the area was a perfect circle.

## Analysis of MSI data

The mass spectra obtained in each probe scan were merged into a single imaging data file using the Perl script provided by Waters. The imaging data were processed via HDI imaging (ver. 1.6, Waters) for peak detection. The

output data were converted into imzml files. The files were then converted to the IMDX format by an IMDX converter (Shimadzu). Data analysis was conducted via IMAGEREVEAL (Shimadzu). The regions of interest (ROIs) for each cell were manually set by comparing the fluorescence images and ion images of $m/z$ 760.585. Data preprocessing was performed by normalizing the signal intensities to the total ion count of the mass spectrum for each pixel. Peak detection was conducted for the entire $m/z$ range of 600–1000. The list of ion peaks was used for level 2 putative annotation[77] with the results of LIPID MAPS bulk structure searches[78]. Protonated molecule ($[M + H]^+$), adduct ion with sodium ion ($[M + Na]^+$), and potassium ion ($[M + K]^+$) were the targets, and a mass tolerance of ±0.005 $m/z$ value was used.

Levene's test and Welch's t-test were used to assess significant differences in lipid ion signal intensities between the cell types. The variance between the two groups was examined using Levene's test for signal intensity data of the ROIs. Among the data for the 29 ions, the $p$ values for the 17 ions were below the significance level (0.05). The data for the 17 ions were analyzed using Welch's t-test for two samples with unequal variances. The data for the 12 ions were analyzed using Welch's t-test for two samples with equal variances. The box plots were prepared using DataGraph (Visual Data Tools, USA).

## Lipidomic analysis

Lipid extraction was performed using the Bligh and Dyer method[79] with some modifications. Briefly, HeLa cells (mCAT: $1.8 \times 10^6$ cells, GCS: $1.3 \times 10^6$ cells) were washed twice with 2 mL of PBS and quenched with 1 mL of ice-cold methanol. Each type of cell was cultured separately in three Petri dishes, and samples pooled by cell type were used. After being scraped, the cell suspension (~1 mL) was transferred into a 2 mL Eppendorf tube and mixed with 10 µL of internal standard (IS) solution A (Mouse SPLASH Lipidomix Mass Spec Standard, Avanti Polar Lipids, Inc., USA) containing phosphatidylcholine (PC) $15:0/[^2H_7]18:1$ (1.0 nmol), phosphatidylethanolamine (PE) $15:0/[^2H_7]18:1$ (0.070 nmol), phosphatidylserine (PS) $15:0/[^2H_7]18:1$ (0.20 nmol), phosphatidylglycerol (PG) $15:0/[^2H_7]18:1$ (0.050 nmol), phosphatidylinositol (PI) $15:0/[^2H_7]18:1$ (0.20 nmol), phosphatidic acid (PA) $15:0/[^2H_7]18:1$ (0.10 nmol), lysophosphatidylcholine (LPC) $[^2H_7]18:1$ (0.45 nmol), lysophosphatidylethanolamine (LPE) $[^2H_7]18:1$ (0.020 nmol), cholesteryl ester (ChE) $[^2H_7]18:1$ (2.5 nmol), diacylglycerol (DG) $15:0/[^2H_7]18:1$ (0.15 nmol), triacylglycerol (TG) $15:0/[^2H_7]18:1/15:0$ (0.35 nmol), and sphingomyelin (SM) $d18:1/[^2H_9]18:1$ (0.20 nmol) and 10 µL of IS solution B (Avanti Polar Lipids Inc.) containing ceramide (Cer) $d[^2H_7]18:1/15:0$ (0.10 nmol), hexosylceramide (HexCer) $d[^2H_5]18:1/18:1$ (0.10 nmol), free fatty acid (FA) $[^{13}C_{16}]16:0$ (0.10 nmol), monoacylglycerol (MG) $[^2H_7]18:1$ (1.0 nmol), and $[^2H_7]$cholesterol (3.0 nmol). The samples were vigorously mixed by vortexing them for 1 min, followed by sonication for 5 min. To precipitate proteins, the solvent extracts were incubated on ice for 5 min. The extracts were then centrifuged at $16,000 \times g$ for 5 min at 4 °C, and the resulting supernatant was transferred to clean tubes. The supernatant (400 µL) was dried under a stream of nitrogen and stored at −80 °C until analysis. Prior to analysis, the dried sample was reconstituted in 100 µL of methanol/chloroform (1:1, v/v). Lipidomic analysis was performed via supercritical fluid chromatography with a diethylamine (DEA) column coupled to an Orbitrap Exploris 120 high-performance benchtop quadrupole Orbitrap mass spectrometer (Thermo Fisher Scientific, USA) (SFC-MS/MS)[80,81].

The SFC (Nexera UC system, Shimadzu) conditions were as follows: column, ACQUITY UPC$^2$ Torus DEA column (3.0 mm i.d. × 100 mm, 1.7 µm particle size, Waters); injection volume, 2 µL; column temperature, 50 °C; mobile phase A, supercritical carbon dioxide (99.9%, Yoshida Sanso, Japan); mobile phase B (modifier) and make-up pump solvent, methanol/water (95/5, v/v) with 0.1% (w/v) ammonium acetate; flow rate of the mobile phase, 1.0 mL/min; flow rate of the make-up pump, 0.1 mL/min; and pressure of the back pressure regulator, 10 MPa. The gradient conditions were as follows: 1% B, 0–1 min; 1–75% B, 1–24 min; 75% B, 24–26 min; and 1% B, 26–30 min.

The full-scan MS analysis conditions were as follows: polarity, positive and negative ionization with a positive/negative fast polarity switching function; sheath gas flow rate, 10 arb for positive ionization and 50 arb for negative ionization; auxiliary (Aux) gas flow rate, 0 arb for positive ionization and 10 arb for negative ionization; spray voltage, 3.0 kV for positive ionization and $-2.0$ kV for negative ionization; ion transfer tube temperature, 320 °C; S-lens level, 60; vaporizer temperature, 100 °C; mass resolution, 60,000; automatic gain control (AGC) target (number of ions to fill C traps), $1 \times 10^6$; maximum injection time (IT), 200 ms; and scan range, 200–2000 ($m/z$). The conditions for MS/MS using parallel reaction monitoring for each target compound were as follows: mass resolution, 15,000; AGC target, $1 \times 10^6$; maximum IT, 80 ms; isolation window, 0.4 Da; RF-lens; 60%; fixed first mass, 70 $m/z$; collision energy type, normalized; and collision energy, 10–30%. The details are shown in Supplementary Table 5. The analytical platform for lipidomic analysis was controlled via LabSolutions (version 5.99 SP2; Shimadzu Co.) and Xcalibur (version 4.3; Thermo Fisher Scientific). Data analysis was performed via Multi-ChromatoAnalysT (version 1.3.4.0; Beforce, Japan). The lipids were identified based on the retention time, precursor ion, and fragmentation patterns of each molecule. The quantitative levels of lipids were calculated from the peak areas relative to those of the IS.

## Data availability
The MSI data included in this study are available from the corresponding author upon request.

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

## Acknowledgements

We sincerely thank Dr. Yasushi Ishihama and Dr. Naoyuki Sugiyama of Kyoto University for their valuable advice on the preparation of the capillary probes. We sincerely thank Dr. Thanai Paxton of the Waters Corporation for his support regarding the use of the mass spectrometer and data analysis. We sincerely thank Dr. Toshiyuki Yamaji of the National Institute of Infectious Diseases in Japan for providing the HeLa mCAT and HeLa IP-cGCS cell lines. We thank Ms. Maiko Goto for her help with the lipidome analysis. This work was partly supported by JST, PRESTO Grant Number JPMJPR20E4, a Grant-in-Aid for Scientific Research on Innovative Areas "Chemistry for Multimolecular Crowding Biosystems" (JSPS KAKENHI Grant No. JP20H04710), JSPS KAKENHI Grant No. JP23H03711, JP23K17880, the Asahi Glass Foundation, Basis for Supporting Innovative Drug Discovery and Life Science Research (BINDS) [JP23ama121055] from AMED. This work was also supported in part by the MEXT Cooperative Research Project Program, Medical Research Center Initiative for High Depth Omics, and CURE:JPMXP1323015486 for the Medical Institute of Bioregulation (MIB), Kyushu University.

## Author contributions

Y.O. conducted project design, funding, instrument development, and experimental data acquisition and analysis. K.K. and A.M. cultured HeLa cells and worked with Y.O. on pre-treatment methods. M.T., K.H., and Y.I. analyzed the lipids in HeLa cells provided by K.K. and A.M. using SFC-MS/MS. T.B., K.F., and M.T. provided advice to facilitate the study. Y.O. wrote the first draft of the manuscript and prepared figures. All authors have reviewed the results and approved the final version of the manuscript.

## Competing interests

The authors declare no competing interests.
