## [Transparent Peer Review file · Communications Chemistry]

Single-Cell Mass Spectrometry Imaging of Lipids in HeLa Cells via Tapping-Mode Scanning Probe Electrospray Ionization

Corresponding Author: Dr Yoichi Otsuka

Version 0:

Reviewer comments:

Reviewer #1

(Remarks to the Author)

The manuscript of Otsuka et al presents an interesting and relevant study combining single-cell mass spectrometry imaging (SC-MSI) with tapping-mode scanning probe electrospray ionization (t-SPESI) for lipid analysis. The integration of SC-MSI with microscopy provides valuable insights, particularly in distinguishing between two HeLa cell lines based on lipid ion signal intensities. The approach is innovative and holds promise for advancing lipidomic analysis at the single-cell level. Below, I provide minor suggestions for improvement, along with requests for clarification to enhance the overall clarity and impact of the manuscript.

Specific Comments and Questions

1. Lines 151–152:

“The solvent was supplied from a nanoflow pump to the probe. When the probe tip contacted the sample, a liquid bridge formed between them and the sample components were extracted.”

o How is uniform extraction ensured? Could the researchers clarify whether differences in extraction volume are normalized, for instance, by introducing internal standards into the infusion solvent?

2. Line 182:

“The probe position could be dynamically adjusted to account for changes in the sample height in the range of 10 μm .”

o The adjustment for nuclear height is commendable, but would this method be feasible for analyzing dividing cells?

Additionally, are there potential limitations in achieving equitable extraction between cytoplasm and nucleus?

o One suggestion would be to fix the analyzed specimens and homogeneously infuse an internal standard to demonstrate uniform extraction across different heights and cellular compartments. Could the authors comment on this or propose alternative approaches to ensure accurate signal normalization?

3. General Query:

o Could authors please state clearly in the results section if the cells fixed, dead, or maintained in growth media during analysis? If in growth media, could the authors comment on the potential contribution of media components to background ions?

4. Lines 296–297:

“Because the ion signal intensity in this measurement system is currently insufficient for tandem mass spectrometry (MS/MS), lipidomic analysis via SFC-MS/MS was performed to evaluate the lipids in HeLa cells.”

o The description of lipidomic analysis in the results section requires further clarification. For instance, how many cells were analyzed in bulk? Including these details would improve the manuscript’s reproducibility and transparency.

5. Lines 302–303:

“No obvious differences were observed in the overall mass spectrum patterns; thus, the signal intensities of the individual ion peaks were compared.”

o How can the authors ensure that observed intensity differences between the cell lines are not influenced by ion suppression effects?

o Testing for ion suppression using standards, such as Avanti SPLASH mix, infused into the solvent flow (e.g., methanol) might strengthen the study. Could the researchers discuss additional alternatives or experimental strategies to address this?

General Suggestions:

• To further demonstrate the method's utility, could the researchers use the genetic differences between the two cell lines to perform a perturbation experiment? For example, introducing redox stress could highlight the heterogeneity in metabolic responses between the cell lines, aligning with their distinct genetic backgrounds.

• As an additional control, tagging one of the cell lines with GFP and mixing both cell lines on the same slide could mitigate batch effects and external contaminants. Could the researchers test whether the MS data successfully differentiates the mixed cells?

Overall, this is a compelling manuscript with significant potential. Addressing the above points would enhance its clarity, rigor, and impact.

Reviewer #2

(Remarks to the Author)

Authors reported studies of developing the tapping-mode scanning probe electrospray ionization (t-SPESI) for single-cell mass spectrometry imaging (SC-MSI) of cells. The t-SPESI unit was used to extract components from microregions of cells for MS analysis. The t-SPESI unit was combined with an inverted fluorescence microscope for multimodal (fluorescence, lipid ion, and topographic) imaging of cells labeled with fluorescent dyes. The technical allowed for distinguishing two cell types as well as investigating differences in the subcellular localization of lipids in cells. In general, authors carefully designed a sophisticated setup to study single cells. Although this system has the potential to image subcellular distributions of molecules, its capabilities of subcellular analysis are limited.

Major concerns:

- (1) The subcellular abundances of molecules are proportional to the total amounts of cellular analytes, whereas limited information of cell functions can be obtained. According to Fig. 5, the relative ion abundances are inversely proportional to the distribution of cell nucleus (or heights), where cellular contents are less compared with those regions. This type of information can be directly obtained from fluorescence microscopy, whereas the MSI studies seem very necessary. It is unclear if (and how) the ion intensity normalization was performed. More careful data analysis may provide detailed information.
- (2) It is unclear if the oscillation frequency was optimized. With a frequency of 656-657 Hz, the contact time (or extraction time) with cell surface is very short, and this may affect the extraction of cellular contents in deeper regions in cells, limiting its capability of subcellular analysis.
- (3) Some designs can be simplified. Authors packed silica beads into the capillary probe to minimize clogging issues. This seems a little too complicated. They could filter solvents before experiments. In addition, an in-line microfilter or LC trap column, which are commercially available, can be added between the solvent pump and probe. In fact, adding an LC trap column can provide certain back pressure to stabilize solvent flow at very low flowrate.
- (4) Some explanations need to be corrected.
 - (a) In line 83-85, mechanisms of MALDI need to be corrected. Although the exact mechanisms of MALDI are not clearly understood, it is generally agreed that the matrix molecules are desorbed and excited to their electronically excited states to facilitate proton transfer. The statement "The energy of the laser beam is converted to thermal energy, resulting in desorption and ionization of the sample components." is inaccurate, because photon energy is partially converted to kinetic energy (desorption), but the majority is deposited into electronically excited states to facilitate ionization processes.
 - (b) In line 251-252, it says "increasing temperature can be attributed to both the enhanced desolvation of the charged droplets and the change in the gas flow inside the transfer tube from turbulent to laminar." In fact, increasing temperature decreases viscosity, resulting in a transition from laminar to turbulent flow. Thus, the major mechanism is the enhanced desolvation.
 - (c) The function of S-type tube was not clearly addressed. It is very likely that S-shaped tube can effectively remove large droplets (neutral or charged) and enhance desolvation (more collisions with tube surface) and ionization efficiency.

Minor suggestions:

- (1) More recent reviews of single cell MS techniques, particularly ambient methods, can be cited in the introduction such as "Single Cell mass spectrometry: Towards quantification of small molecules in individual cells" *TrAC Trends in Analytical Chemistry*, 2024, Volume 174, 117657, 10.1016/j.trac.2024.117657 and "Recent Developments in Single-Cell Metabolomics by Mass Spectrometry—A Perspective", <https://doi.org/10.1021/acs.jproteome.4c00646>
- (2) It is unclear why Welch's t-test (line 305) was used. Did authors conduct Levene's test to determine if the variances of two groups are equal?
- (3) Authors need to check the solvent flowrate (line 479). They stated that the flowrate is nano 1 nL/min by using LC-AD20 nano LC pump. In fact, the lowest flowrate that can be provided by this model of pump is 0.1 uL/min.

Reviewer #3

(Remarks to the Author)

The authors describe work performed to image lipids in cells on a surface using a liquid extraction technique, t-SPESI, coupled to mass spectrometry. The manuscript is interesting, showcases the technique and report novel images using the technique. The manuscript indicates that several aspects of the development have been taken into consideration and optimized, which makes it seem non-focused and lacking detailed information. The major claims are the optimization and use of t-SPESI for subcellular imaging of cells. However, there is a clear lack of data to support observations and speculations. Overall, the manuscript has the potential to be important in the field of single cell mass spectrometry/lipidomics after including data and being thoroughly rewritten.

The introduction indicates a focus on cancer that is not followed up in the manuscript. The wording is sometimes unconventional, such as significant imaging technique, ionization region, metabolic transformation, irradiation of primary ion beam. Additionally, the introduction includes many details on pixel sizes for various techniques except for ToF-SIMS. Thus, the introduction need to be carefully edited.

The start of the results and discussion section with an overview is redundant since it does not provide the reader with enough information to be interesting. In several places the authors discuss the "previous system" in detail. This should be moved around to instead focus on the system that is being described and the importance of the updates. The addition of particles into the probe should be reduced and instead detailed in the SI, including data on improved performance. Contrarily, there should be more details on the results shown in the main figures, including the importance of the height shifts.

What tolerance does the system have for height variation? How is the height calibrated? From SI fig 2 it seems like the distance stated as 1 μm step is a 50 μm step. Furthermore, the b figure stating ~ 0.3 V for 1 μm is very different from fig a stating ~ 7 V for the same step. This need to be clarified.

How is the distance and amplitude validated? Evaluate and show data.

When altering the probe distance, this should also alter the distance to the inlet of the MS. Does this result in different signal intensities? When does the distance start changing the signal? Please elaborate and show data.

Is the technique really so sensitive to topography that less than 1 μm height difference matter? What are the limitations? How is the topography calibrated? Please add data.

Is there a way to get around shadowing during imaging?

What solvent is used for the measurements? Does the solvent matter?

How is the size of the liquid bridge measured? Please add data showing potential deviations from the stated 2.3 μm .

How does the frequency correlate with the size of the liquid bridge? How does evaporation impact the measurements? Show data.

How is the pixel size determined? How is it set? Please add data to validate the results.

The delay in feedback is stated to be minimal. Please add data to show this.

Why is NaI used for optimization? And why is not the results from this used for the imaging of the cells? Including flow rate and temperature. Differences in applied voltage, use of from 300 nL/min to 1 nL/min, and 50 degrees difference in temperature suggests that the optimization is not used. Please optimize using metabolites in the same solvent and use the optimal settings for collecting the data.

The observed improvement using the smaller tube need to be confirmed with data. The speculations to why these observations occur need to be substantiated.

What is the patterned substrate used for the cells? In addition to the observataion, show data that this matters.

How is the chemical and spatial integrity of the cells preserved? What does the staining and fixation do to the mass spectrum and contaminations? Are the contaminations close to the m/z of the reported lipids? Please show data showing mass spectrum of only cells, only staining, only fixation and the combination of the three.

How do you know that the cells are not overlapping on the surface?

It does not seem like the ROI is of 45 cells since less than this are shown in the figures.

Please add the mass resolving power of the mass spectrometer used to allow for a mass tolerance of 0.005 Da. Only high res qtof can provide data with the three decimals used in the manuscript. Annotation on low res MS data is not recommended.

How was the significance calculated between the two cell types? P-values? Please specify.

How many endogenous molecules was detected from the cells? The manuscript only states small amount.

The meaning of the distribution of lipids in the cells seem highly overstated. There are no lipids inside the nucleus, which is clear in the PC images. This makes me think that the other images show contaminations and noise and not distributions of endogenous lipids.

The experiment to try to correlate quantification with SFC and t-SPESI is not understandable and needs to be revised.

How was the scanning speed of the probe optimized to the very specifi 6.7 $\mu\text{m}/\text{s}$? Please show data.

Version 1:

Reviewer comments:

Reviewer #1

(Remarks to the Author)

Thank you for addressing my comments clearly. Indeed, quantitation will remain a future challenge and ion suppression effects will need to be addressed to be able to successfully extract biological insights from this (and related) technologies.

Reviewer #2

(Remarks to the Author)

Authors have addressed all comments. There are no additional comments.

Dear Reviewers,

We would like to express our sincere gratitude for your careful review and thoughtful comments regarding our manuscript.

Your insightful feedback has been invaluable in helping us refine and improve the quality of our manuscript. We deeply appreciate the time and effort you have devoted to evaluating our research and we are truly grateful for your constructive suggestions.

We have provided point-by-point responses to each of the reviewers' comments.

We would be truly grateful if you could kindly review our revised manuscript and provide further insights, if necessary.

Sincerely,

Yoichi Otsuka

Responses to the reviewers' comments

Reviewer #1

The manuscript of Otsuka et al presents an interesting and relevant study combining single-cell mass spectrometry imaging (SC-MSI) with tapping-mode scanning probe electrospray ionization (t-SPESI) for lipid analysis. The integration of SC-MSI with microscopy provides valuable insights, particularly in distinguishing between two HeLa cell lines based on lipid ion signal intensities. The approach is innovative and holds promise for advancing lipidomic analysis at the single-cell level. Below, I provide minor suggestions for improvement, along with requests for clarification to enhance the overall clarity and impact of the manuscript.

Specific Comments and Questions

Q1-1.

Lines 151–152:

“The solvent was supplied from a nanoflow pump to the probe. When the probe tip contacted the sample, a liquid bridge formed between them and the sample components were extracted.”

o How is uniform extraction ensured? Could the researchers clarify whether differences in extraction volume are normalized, for instance, by introducing internal standards into the infusion solvent?

A1-1.

To realize SC-MSI using t-SPESI, elemental technologies to reduce the size of the liquid bridge are necessary. We used the measurement system combined with an inverted microscope to directly observe the liquid bridge from the backside while performing the MSI. We confirmed that the solvent did not spread significantly over the glass substrate or cells during probe scanning. To the best of our knowledge, there have been no reports of techniques for observing the extraction area from the backside of the sample using other direct extraction ionization methods such as DESI and nano-DESI.

In addition, syringe pumps are often used for solvent flow, but the issue with syringe pumps is that it is difficult to monitor the pressure and flow rate of the solvent. Therefore, we used a nanoflow pump to confirm that the solvent flow rate and pressure were maintained while performing MSI.

We considered that the formation of a uniform extraction area on a microscale was achieved by combining both direct observation of the liquid bridge using a microscope and monitoring the status of the nanoflow pump.

Your suggestion to estimate the extracted area by adding an internal standard to the solvent is conceivable; however, it would be difficult to determine the variation in the amount of solvent used for extraction because ionization suppression occurs when measuring biological samples.

<Added>

Page 13, line 202 of the marked manuscript.

To realize SC-MSI using t-SPESI, elemental technologies to reduce the size of the liquid bridge are necessary. We used the measurement system combined with an inverted microscope to directly observe the liquid bridge from the backside while performing the MSI. We confirmed that the solvent did not spread significantly over the glass substrate or cells during probe scanning. To the best of our knowledge, there have been no reports of techniques for observing the extraction area from the backside of the sample using other direct extraction ionization methods, such as DESI and nano-DESI, during MSI. In addition, syringe pumps are often used for solvent flow, but the issue with syringe pumps is that it is difficult to monitor the pressure and flow rate of the solvent. Therefore, we used a nanoflow pump to confirm that the solvent flow rate and pressure were maintained while performing MSI. We considered that microscale extraction could be achieved by combining both direct observation of the liquid bridge using a microscope and monitoring the status of the nanoflow pump.

Q1-2.

Line 182:

“The probe position could be dynamically adjusted to account for changes in the sample height in the range of 10 μm .”

o The adjustment for nuclear height is commendable, but would this method be feasible for analyzing dividing cells? Additionally, are there potential limitations in achieving equitable extraction between cytoplasm and nucleus?

A1-2.

We defined your expression “equitable extraction” as “extraction using a constant volume of solvent”. When measuring real samples with uneven surfaces, it is important to make the contact between the probe and sample uniform by making the probe oscillation amplitude constant. The newly developed measurement system is equipped with a feedback control mechanism.

As shown in Fig. 4, we confirmed that the oscillation amplitude remained within 1% even when the sample height varied by up to 1 μm . If the sample height variation is within the movable range of the piezo stage that holds the t-SPESI unit, it is conceivable that the dividing cells can be measured.

In addition, since the oscillation amplitude was maintained constant, it is thought that extraction and ionization using a constant amount of solvent was carried out in different regions of the cell nucleus and cytoplasm.

The results in Fig. 4 show that the amplitude slightly varied by approximately ± 1 mV, especially due to the high height of the nucleus; however, we considered that this was not a major barrier to obtaining the lipid distribution of the cell. If the feedback control had a negative effect, the signal intensity would be expected to vary depending on the relative position of the nucleus and cytoplasm; however, similar ion images were obtained for each cell, even when the position of the cytoplasm was different from that of the nucleus (cytoplasm above or below the nucleus, or to the left or right of the nucleus). For example, in the ion image of PC 34:1 [M+H]⁺, the signal intensity of the cell nucleus was lower and the intensity of the outer part of the nucleus was higher.

<Added>

Page 19, line 317 of the marked manuscript.

If the feedback control had a negative effect, the signal intensity would be expected to vary depending on the relative position of the nucleus and cytoplasm; however, similar ion images were obtained for each cell. For example, in Fig. 4b and 4f, the signal intensities of the cell nuclei were lower and the intensities of the cytoplasm were higher.

Q1-3.

o One suggestion would be to fix the analyzed specimens and homogeneously infuse an internal standard to demonstrate uniform extraction across different heights and cellular compartments. Could the authors comment on this or propose alternative approaches to ensure accurate signal normalization?

A1-3.

As with other extraction-ionization methods, the challenge with t-SPESI is that ion suppression makes it difficult to normalize the mass spectrum accurately. As you suggest, fixing the analyte and normalizing using an internal standard is conceivable as a way to achieve a certain level of normalization.

In contrast, cells contain a complex molecular crowding environment, and when targeting multiple lipids with structural diversity, there is a possibility of them being affected by ion suppression. Therefore, in this paper, we did not use internal standards.

To ensure accurate signal normalization, it is necessary to use a sample that is chemically and morphologically well-defined, rather than a complex sample with multiple components, such as cells or biological tissue.

For example, we think it would be necessary to use standard samples in which multiple known biological components are applied uniformly to a substrate with a known uneven shape. There is no doubt that such samples are important not only for the normalization of MSI data but also for examining ionization efficiency. Unfortunately, a specific preparation technique has not yet been established. In addition, it is necessary to consider a data analysis method that would ensure accurate signal normalization by comprehensively evaluating the topography, amplitude images, and ion images using a solvent to which an internal standard has been added.

<Added>

Page 24, line 405 of the marked manuscript.

To ensure accurate signal normalization, it is necessary to use a sample that is chemically and morphologically well-defined, rather than a complex sample with multiple components, such as cells or biological tissue. Furthermore, it is necessary to consider a data analysis method that comprehensively evaluates the topography, amplitude image, and ion image using a solvent to which an internal standard has been added.

Q1-4.

3. General Query:

o Could authors please state clearly in the results section if the cells fixed, dead, or maintained in growth media during analysis? If in growth media, could the authors comment on the potential contribution of media components to background ions?

A1-4.

We referred to papers from other research groups for the sample preparation (Ref #75). After staining the cells cultured on the glass substrate with fluorescent dye, we performed chemical fixation and rinsing to wash out the components contained in the culture medium.

<Added>

Page 10, line 152 of the marked manuscript.

The cells were stained with fluorescent dyes, chemically fixed, and dried.

Q1-5.

Lines 296–297:

“Because the ion signal intensity in this measurement system is currently insufficient for tandem mass spectrometry (MS/MS), lipidomic analysis via SFC-MS/MS was performed to evaluate the lipids in HeLa cells.”

o The description of lipidomic analysis in the results section requires further clarification. For instance, how many cells were analyzed in bulk? Including these details would improve the manuscript's reproducibility and transparency.

A1-5.

In the SFC-MS/MS analysis, 1.8×10^6 cells of mCAT cells and 1.3×10^6 cells of GCS cells were analyzed. Each type of cell was cultured separately in three Petri dishes, and samples pooled by cell type were used.

<Added>

Page 35, line 596 of the marked manuscript.

Each type of cell was cultured separately in three Petri dishes, and samples pooled by cell type were used.

Q1-6.

Lines 302–303:

“No obvious differences were observed in the overall mass spectrum patterns; thus, the signal intensities of the individual ion peaks were compared.”

o How can the authors ensure that observed intensity differences between the cell lines are not influenced by ion suppression effects?

o Testing for ion suppression using standards, such as Avanti SPLASH mix, infused into the solvent flow (e.g., methanol) might strengthen the study. Could the researchers discuss additional alternatives or experimental strategies to address this?

A1-6.

This is related to question 2, and it is difficult to prove that ion signal intensities are not affected by ion suppression. Without chromatography, ion suppression is a common problem in many ionization methods, such as MALDI, DESI, and nano-DESI, and remains an unsolved research topic.

In this study, we showed that the correlation between the signal intensity measured by t-SPESI and the amount of lipids measured by SFC-MS/MS was positive or negative, depending on the lipid type (Supplementary Fig. 10). This result indicates that ion suppression occurred.

We also agree that adding an internal standard to the solvent may be useful for normalizing the signal intensity of MSI data. In contrast, the nanoflow pump used in this study uses a split-flow system. Although a portion of the solvent was supplied to the t-SPESI probe, the majority was refluxed to the solvent bottle. The cap of the solvent bottle contains a gap for inserting the tube for reflux, and it is conceivable that the solvent will gradually evaporate from this point. This

could cause changes in the concentration of the internal standard, and there is a concern that this could affect the normalization of the mass spectrum.

What we want to present in this paper is that we have experimentally shown that the signal intensity of lipids in cells changes relatively depending on the cell type in t-SPESI-MSI. The use of internal standards is a topic of future research.

Q1-7.

General Suggestions:

- To further demonstrate the method's utility, could the researchers use the genetic differences between the two cell lines to perform a perturbation experiment? For example, introducing redox stress could highlight the heterogeneity in metabolic responses between the cell lines, aligning with their distinct genetic backgrounds.
- As an additional control, tagging one of the cell lines with GFP and mixing both cell lines on the same slide could mitigate batch effects and external contaminants. Could the researchers test whether the MS data successfully differentiates the mixed cells?

A1-7.

Thank you for your thought-provoking comments. We are continuing to develop analytical techniques so that this technology can become a helpful tool in the field of cell biology. The greatest feature of single-cell MSI is that it can visualize the localization of multiple molecules in individual cells, which is difficult to do with subcellular imaging using a fluorescence microscope or cell analysis using flow cytometry.

In this study, we imaged and compared two different types of genetically regulated cells. As you pointed out, we found the themes of analyzing changes in metabolism when cells were given external stimuli and identifying multiple cells in a single field of view to be interesting. As we specialize in the development of analytical techniques, we would like to take the opportunity to collaborate with researchers who use such cell samples.

<Added>

Page 25, line 426 of the marked manuscript.

Compared with conventional MSI, SC-MSI is expected to significantly improve the quality of spatial omics information of biological tissues and complement imaging using fluorescence microscopy and cell analysis using flow cytometry. The evolution of SC-MSI, which can be used to measure the multidimensional molecular information of individual cells that make up diseased tissue and/or have been subjected to external stress, is expected to contribute to the derivation of new knowledge for biomarker discovery, elucidation of pathogenesis, and heterogeneity in metabolic responses.

Q1-8.

Overall, this is a compelling manuscript with significant potential. Addressing the above points would enhance its clarity, rigor, and impact.

A1-8.

We sincerely appreciate your thorough and insightful comments. Your constructive comments and suggestions are invaluable for enhancing the clarity, rigor, and impact of our manuscript. Thank you for your time and effort in evaluating our manuscript.

Reviewer #2:

Authors reported studies of developing the tapping-mode scanning probe electrospray ionization (t-SPESI) for single-cell mass spectrometry imaging (SC-MSI) of cells. The t-SPESI unit was used to extract components from microregions of cells for MS analysis. The t-SPESI unit was combined with an inverted fluorescence microscope for multimodal (fluorescence, lipid ion, and topographic) imaging of cells labeled with fluorescent dyes. The technical allowed for distinguishing two cell types as well as investigating differences in the subcellular localization of lipids in cells. In general, authors carefully designed a sophisticated setup to study single cells. Although this system has the potential to image subcellular distributions of molecules, its capabilities of subcellular analysis are limited.

Q2-1.

Major concerns:

(1) The subcellular abundances of molecules are proportional to the total amounts of cellular analytes, whereas limited information of cell functions can be obtained. According to Fig. 5, the relative ion abundances are inversely proportional to the distribution of cell nucleus (or heights), where cellular contents are less compared with those regions. This type of information can be directly obtained from fluorescence microscopy, whereas the MSI studies seem very necessary. It is unclear if (and how) the ion intensity normalization was performed. More careful data analysis may provide detailed information.

A2-1.

We normalized the mass spectra of each pixel of the ion image by the total ion signal intensity (as described in the Analysis of MSI data section of the main text). Even without normalization, a decrease in the lipid signal intensity in the nucleus was observed. Our results show that the amount of lipids in the nucleus is low and qualitatively consistent with the results of other imaging techniques.

Phospholipids in the nucleus are mainly present in the nuclear membrane, and it has been reported that in rat liver cells, the main components inside the nucleus are proteins and DNA (Robert W. Ledeen, Gusheng Wu, Nuclear lipids: key signaling effectors in the nervous system and other tissues, *Journal of Lipid Research*, Volume 45, Issue 1, 2004, Pages 1-8).

The results of single-cell MSI using MALDI have reported that the signal intensity of phospholipids decreases in the region of the nucleus of Vero B cells (Ref #37, Niehaus, M., Soltwisch, J., Belov, M.E. et al. Transmission-mode MALDI-2 mass spectrometry imaging of cells and tissues at subcellular resolution. *Nat Methods* 16, 925–931 (2019).).

In addition, a decrease in Raman signals derived from lipids in the cell nucleus has been reported in observations of HeLa cells using stimulated Raman scattering microscopy (Wilson, Liam T. and Tipping, William J. and Wetherill, Corinna and Henley, Zoë and Faulds, Karen and Graham, Duncan and Mackay, Simon P. and Tomkinson, Nicholas C. O., *Mitokyne: A Ratiometric Raman Probe for Mitochondrial pH*, *Analytical Chemistry*, 93, 12786-12792, (2021).).

Gene expression is regulated in the nucleus. Therefore, it is conceivable that the mass spectrometry of molecules in the nucleus is important. The investigation of the solvent conditions for the selective ionization of components localized in the nucleus is a future challenge.

<Added>

Page 22, line 370 of the marked manuscript.

The presented results show that the amount of lipids in the nucleus is low and qualitatively consistent with the results of other imaging techniques. Phospholipids in the nucleus are mainly present in the nuclear membrane, and it has been reported that in rat liver cells, the main components inside the nucleus are proteins and DNA (Ref #67). The results of single-cell MSI using MALDI have shown that the signal intensity of phospholipids decreases in the nucleus of Vero B cells (Ref #37). In addition, a decrease in Raman signals derived from lipids in the cell nucleus has been reported in observations of HeLa cells using stimulated Raman scattering microscopy (Ref #68).

Q2-2.

(2) It is unclear if the oscillation frequency was optimized. With a frequency of 656-657 Hz, the contact time (or extraction time) with cell surface is very short, and this may affect the extraction of cellular contents in deeper regions in cells, limiting its capability of subcellular analysis.

A2-2.

In this paper, we emphasize that SC-MSI can be performed by suppressing solvent spreading using t-SPESI, which repeats the extraction and ionization with small amounts of solvent at high speed. Under the probe oscillation conditions used in this study, it was estimated that the

extraction-ionization process was repeated on average 197 times to acquire the mass spectrum for one pixel in the MSI. In addition, it was confirmed that the trace of probe scanning remained after the measurement (Supplementary Fig.11). Thus, we consider that the probe penetrated the interior of the sample and performed extraction and ionization.

Optimization of the oscillation frequencies would be important for improving the spatial resolution and increasing the ion signal intensity. To improve the spatial resolution, it is necessary to use a capillary probe with a finer tip and further reduce the size of the liquid bridge. However, as the solvent volume decreased, the number of extracted molecules decreased. One conceivable method to overcome this problem is to increase the oscillation frequencies for the extraction-ionization event, thereby preventing a decrease in the ion detection sensitivity. In t-SPESI, the resonance frequency of the capillary probe is proportional to the reciprocal of the square of the probe length (Ref #52). Therefore, the resonance frequency can be increased by reducing the probe length. In the future, we plan to use the measurement system presented in this paper to study the relationship between the probe oscillation frequencies, size of the liquid bridge, and mass spectra.

<Added>

Page 24, line 410 of the marked manuscript.

Under the probe oscillation conditions used in this study, it was confirmed that the trace of probe scanning remained after measurement (Supplementary Fig.11). Thus, we consider that the probe penetrated the interior of the sample and performed extraction and ionization. Optimization of the oscillation frequencies is important for improving the spatial resolution and increasing the ion signal intensity. To improve the spatial resolution, it is necessary to use a capillary probe with a finer tip and further reduce the size of the liquid bridge. However, as the solvent volume decreased, the number of extracted molecules decreased. One conceivable method to overcome this problem is to increase the oscillation frequencies for the extraction-ionization event, thereby preventing a decrease in the ion detection sensitivity. In t-SPESI, the resonance frequency of the capillary probe is proportional to the reciprocal of the square of the probe length (Ref #52). Therefore, the resonance frequency can be increased by reducing the probe length. In the future, we plan to use the measurement system presented in this paper to study the relationship between the probe oscillation frequencies, size of the liquid bridge, and mass spectra.

Q2-3.

(3) Some designs can be simplified. Authors packed silica beads into the capillary probe to minimize clogging issues. This seems a little too complicated. They could filter solvents before experiments. In addition, an in-line microfilter or LC trap column, which are commercially

available, can be added between the solvent pump and probe. In fact, adding an LC trap column can provide certain back pressure to stabilize solvent flow at very low flowrate.

A2-3.

Thank you for your comments. To perform single-cell MSI using t-SPESI, it is necessary to reduce the size of the capillary probe tip to reduce the size of the liquid bridge. We have been working on the clogging problem of capillary probes for many years. As you suggested, the use of inline microfilters and LC trap columns is conceivable to remove impurities from the solvent. However, this was insufficient. In t-SPESI, the major problems were the friction between the sleeve and capillary probe and contamination of the flow path by dust in the air when the capillary probe was attached to the flow path.

In another extraction-ionization method, Nano-DESI, a method of using a laser puller to sharpen both ends of a single silica capillary was proposed (Ref #50, Yin, R., Burnum-Johnson, K.E., Sun, X. et al. High spatial resolution imaging of biological tissues using nanospray desorption electrospray ionization mass spectrometry. Nat Protoc 14, 3445–3470 (2019).). We initially adopted this method, but we found it difficult to process both ends of a single capillary with good reproducibility, and there was also an incident in which fine particles in the solvent blocked the capillary probe on the upstream side; therefore, we sought another method. By filling the beads, we were able to prevent unwanted particles larger than the tip opening system from reaching the tip of the capillary probe and were able to stabilize the solvent flow for long periods of time.

Q2-4.

(4) Some explanations need to be corrected.

(a) In line 83-85, mechanisms of MALDI need to be corrected. Although the exact mechanisms of MALDI are not clearly understood, it is generally agreed that the matrix molecules are desorbed and excited to their electronically excited states to facilitate proton transfer. The statement “The energy of the laser beam is converted to thermal energy, resulting in desorption and ionization of the sample components.” is inaccurate, because photon energy is partially converted to kinetic energy (desorption), but the majority is deposited into electronically excited states to facilitate ionization processes.

A2-4.

Thank you for this suggestion. The explanation of the ionization mechanism of MALDI has been revised as follows.

<Revised>

Page 6, line 88 of the marked manuscript.

Before:

In matrix-assisted laser desorption/ionization (MALDI), a sample with an organic matrix coated on its surface is irradiated with a focused laser beam. The energy of the laser beam is converted to thermal energy, resulting in desorption and ionization of the sample components.

After:

In matrix-assisted laser desorption/ionization (MALDI), when a mixture of matrix and sample is irradiated with a focused ultraviolet pulsed laser beam, a portion of the photon energy is converted into kinetic energy and a desorption occurs. The matrix is also excited electronically to facilitate the ionization processes.

Q2-5.

(b) In line 251-252, it says “increasing temperature can be attributed to both the enhanced desolvation of the charged droplets and the change in the gas flow inside the transfer tube from turbulent to laminar.” In fact, increasing temperature decreases viscosity, resulting in a transition from laminar to turbulent flow. Thus, the major mechanism is the enhanced desolvation.

A2-5.

Thank you for your comments. We agree that the main factor in the increase in the ion signal intensity with increasing temperature is enhanced desolvation. The following text has been revised.

<Revised>

Page 16, line 273 of the marked manuscript.

Before:

The increase in the signal intensity with increasing temperature can be attributed to both the enhanced desolvation of the charged droplets and the change in the gas flow inside the transfer tube from turbulent to laminar.

After:

The increase in the signal intensity with increasing temperature up to 250 °C can be mainly attributed to the enhanced desolvation of the charged droplets.

Q2-6.

(c) The function of S-type tube was not clearly addressed. It is very likely that S-shaped tube can effectively remove large droplets (neutral or charged) and enhance desolvation (more collisions with tube surface) and ionization efficiency.

A2-6.

Thank you for this suggestion. We compared L- and S-type ion transfer tubes with different inner diameters. In addition, both ion transfer tubes were bent into an L-shape (corresponding to the S-

shape in your suggestion). We have appended a sentence explaining that bending the ion transfer tube is likely to have resulted in an improvement in the ionization efficiency.

<Added>

Page 17, line 275 of the marked manuscript.

Inside the ion transfer tube, the detection sensitivity of ions is likely to have been improved by effectively removing large droplets and enhancing desolvation by collisions between the charged droplets and the inner wall of the tube.

Q2-7.

Minor suggestions:

(1) More recent reviews of single cell MS techniques, particularly ambient methods, can be cited in the introduction such as “Single Cell mass spectrometry: Towards quantification of small molecules in individual cells” TrAC Trends in Analytical Chemistry, 2024, Volume 174, 117657, 10.1016/j.trac.2024.117657 and “Recent Developments in Single-Cell Metabolomics by Mass Spectrometry—A Perspective”, doi.org/10.1021/acs.jproteome.4c00646

A2-7.

Thank you for providing us with the review information. As the current situation and challenges of SC-MSI/SC-MS are well organized, we have added it to the reference list in the introduction section, and also cited it in the section where we discussed the issues of quantitative analysis.

<Added>

Ref #20, 72

Q2-8.

(2) It is unclear why Welch’s t-test (line 305) was used. Did authors conduct Levene's test to determine if the variances of two groups are equal?

A2-8.

In the first manuscript, Welch’s t-test for unequal variances was performed. As we did not verify whether the data had unequal variance, we performed Levene's test, as suggested. As a result, among the data for the 29 ions considered in this study, the p-value for the 17 ions fell below the significance level (0.05). The data for the 12 ions that did not meet the significance level were considered to have no difference in variance, and a reanalysis of Welch's t-test for two samples with equal variance was performed. Consequently, the p-value of the t-test changed, but the change was minimal, ranging from 0 to 1.3E-5. In Supplementary Table 6, a column showing the results of Levene's test has been added and the results of the reanalysis have been revised.

<Added>

Page 34, line 586 of the marked manuscript.

Levene's test and Welch's t-test were used to assess significant differences in lipid ion signal intensities between the cell types. The variance between the two groups was examined using Levene's test for signal intensity data of the ROIs. Among the data for the 29 ions, the p-values for the 17 ions were below the significance level (0.05). The data for the 17 ions were analyzed using Welch's t-test for two samples with unequal variances. The data for the 12 ions were analyzed using Welch's t-test for two samples with equal variances.

Q2-9.

(3) Authors need to check the solvent flowrate (line 479). They stated that the flowrate is nano 1 nL/min by using LC-AD20 nano LC pump. In fact, the lowest flowrate that can be provided by this model of pump is 0.1 uL/min.

A2-9.

The name of the pump (LC-AD20 nano) was incorrect. The correct name is LC-20AD nano. This pump uses a split-flow system, and it is possible to monitor the flow rate and pressure of the flow path. The flow rate setting range was 0-5000 nL/min, with a setting step of 1 nL/min. LC-20AD is a different pump with a minimum setting flow rate of 0.1 µL/min.

<Corrected>

Page 32, line 556 of the marked manuscript.

Before: LC-AD20 nano

After: LC-20AD nano

Reviewer #3:

The authors describe work performed to image lipids in cells on a surface using a liquid extraction technique, t-SPESI, coupled to mass spectrometry. The manuscript is interesting, showcases the technique and report novel images using the technique. The manuscript indicates that several aspects of the development have been taken into consideration and optimized, which makes it seem non-focused and lacking detailed information. The major claims are the optimization and use of t-SPESI for subcellular imaging of cells. However, there is a clear lack of data to support observations and speculations. Overall, the manuscript has the potential to be important in the field of single cell mass spectrometry/lipidomics after including data and being thoroughly rewritten.

Q3-1.

The introduction indicates a focus on cancer that is not followed up in the manuscript. The wording is sometimes unconventional, such as significant imaging technique, ionization region, metabolic transformation, irradiation of primary ion beam. Additionally, the introduction includes many details on pixel sizes for various techniques except for ToF-SIMS. Thus, the introduction need to be carefully edited.

A3-1.

Thank you for your suggestion. We have revised the text using the English editing service provided by Nature Springer and apologize for the inappropriate use of the technical terms. The following revisions have been made:

	Before	After
1	Compared with imaging techniques such as optical microscopy, this molecular-specific imaging technique is significant .	Compared with imaging technologies such as optical microscopy, MSI has the advantage in that it can obtain multiple molecular-specific images in a single measurement .
2	By miniaturizing the ionization region ,	By reducing the area of the region subjected to ionization on the tissue
3	the metabolic transformation of cells	the changes in metabolism within the cell
4	primary ion beams are irradiated onto a sample	primary ion beam is used to sputter a sample surface

In addition, the pixel size was added to the explanation of SC-MSI using ToF-SIMS.

In the introduction, malignant disease was mentioned as an example of a disease that involves the heterogeneity of cellular networks. This is because HeLa cells were used in this study. As this paper does not cover diseased tissue, it is not necessary to focus on cancerous tissue, as you have suggested. The following references have been added to cover other diseases associated with cellular heterogeneity.

<Revised>

Page 4, line 42 of the marked manuscript.

Before:

Local heterogeneity in the cellular networks that constitute tissues leads to malignant disease¹.

After

Local heterogeneity in the cellular networks that constitute tissues leads to diseases such as malignant tumor (Ref #1), neurodegenerative diseases (Ref #2) and fibrotic diseases (Ref #3).

<Added>

Ref #2

Wilson, D. M., et al. Hallmarks of neurodegenerative diseases. *Cell* **186**, 693-714 (2023).

<https://doi.org/10.1016/j.cell.2022.12.032>

Ref #3

Shaw, T.J. & Rognoni, E. Dissecting Fibroblast Heterogeneity in Health and Fibrotic Disease. *Curr. Rheumatol. Rep.* **22**, 33 (2020).

<https://doi.org/10.1007/s11926-020-00903-w>

Q3-2.

The start of the results and discussion section with an overview is redundant since it does not provide the reader with enough information to be interesting. In several places the authors discuss the “previous system” in detail. This should be moved around to instead focus on the system that is being described and the importance of the updates. The addition of particles into the probe should be reduced and instead detailed in the SI, including data on improved performance. Contrarily, there should be more details on the results shown in the main figures, including the importance of the height shifts.

A3-2.

As you suggested, the explanation of the previous system was redundant. The following text has been deleted.

<Removed>

Page 11, line 174 of the marked manuscript.

In the previous system, the laser source and photodiode unit were placed such that they faced each other. The sample stage size in the horizontal plane was limited to a 30 mm square to prevent interference with the laser source or photodiode unit.

Page 11, line 180 of the marked manuscript.

Similarly to the previous system, a feedback control system for the probe oscillation amplitude is incorporated in the new system. In the previous system, the height of the sample stage was controlled such that the oscillation amplitude was set to a preset value.

In this study, we believe it is necessary to show that the method of packing silica particles into a fused silica capillary is important for performing MSI. To the best of our knowledge, such a capillary probe has never been used in other extraction-ionization methods. We have responded to Reviewer 2's question (3), and the main theme of this paper is the development of a t-SPESI measurement system capable of multimodal imaging and demonstration of single-cell MSI using this system. The method of packing silica particles into a capillary probe is one of the elemental technologies for performing SC-MSI, so we think it is conceivable to explain this in the main text. We have provided a detailed description in Supplementary Fig. 4, which we would like to maintain.

In addition, we have added further details to the results shown in the main figures in accordance with the reviewer's comments. The details have been added to the respective comments, and please refer to them.

Q3-3.

What tolerance does the system have for height variation? How is the height calibrated? From SI fig 2 it seems like the distance stated as 1 μm step is a 50 μm step. Furthermore, the b figure stating ~ 0.3 V for 1 μm is very different from fig a stating ~ 7 V for the same step. This need to be clarified.

A3-3.

We used a step master to calibrate the height information using feedback control. The step master had metal plates of different heights arranged horizontally and the height differences between the metal plates were calibrated. The probe was scanned over four different plates (1, 2, 5, and 10 μm) and the feedback control signals were measured at that time. The profile in Supplementary Fig. 2(a) shows the two metal plates and the gaps between them. The PID output voltage difference for each step was derived by calculating the mean value of the PID output voltage for the plateau portion of each metal plate and taking the difference between the two values.

Supplementary Fig. 2(b) shows the correlation between the step of the metal plate and PID output voltage difference. An equation was obtained to derive the relative height of the sample from the PID output voltage by performing a linear fit of the results of three measurements. The results of this study show that it is possible to measure the relative height of a sample in the range of 1-10 μm by maintaining a constant probe amplitude.

<Corrected>

Page S-3 of the marked manuscript in the supplementary information.

The vertical axis of Supplementary Fig. 2(b) was changed to the PID output voltage difference to match the figure caption.

Q3-4.

How is the distance and amplitude validated? Evaluate and show data.

A3-4.

The oscillation amplitude of the capillary probe was measured by using a laser beam. As the laser beam was exposed between the fixed end and tip of the probe, the measured value was proportional to the displacement of the probe tip.

The oscillation amplitude changes depending on the distance between the capillary probe and sample. The results of an investigation into the relationship between the probe oscillation amplitude and distance are reported. (Mengze Sun, Yoichi Otsuka et al., Probe oscillation control in tapping-mode scanning probe electrospray ionization for stabilization of mass spectrometry imaging. *Analyst* 149, 4011-4019 (2024). <https://doi.org/10.1039/D4AN00712C>)

To measure the actual amplitude of the oscillation, the oscillating probe tip should be brought close to the sample surface. The distance between the probe tip and sample can be measured from the time the probe tip comes into contact with the sample until the probe oscillation stops.

In this study, we obtained SC-MSI data using the same probe. The correlation between the probe oscillation conditions (amplitude and frequency), size of the liquid bridge, and ionization efficiency should be investigated in future studies.

<Added>

Page 19, line 323 of the marked manuscript.

The oscillation amplitude changes depending on the distance between the capillary probe and sample (Ref #65). In this study, SC-MSI data were obtained using the same probe. The correlation between the probe oscillation conditions (amplitude and frequency), size of the liquid bridge, and ionization efficiency should be investigated in future studies.

Q3-5.

When altering the probe distance, this should also alter the distance to the inlet of the MS. Does this result in different signal intensities? When does the distance start changing the signal? Please elaborate and show data.

A3-5.

The inner diameter of the ion transfer tube is 2.18 mm for Type-L and 1.25 mm for Type-S. If the sample has an uneven height of 1 μm , the probe will change in the vertical direction by 1 μm using feedback control. In this situation, the change in height is 0.05% for the Type-L and 0.08% for the Type-S relative to the inlet inner diameter. The ions generated by t-SPESI were aspirated through the large opening of the inlet. Therefore, even if the probe height is changed using feedback control, we believe that the effect on the signal intensity will be small.

Q3-6.

Is the technique really so sensitive to topography that less than 1 μm height difference matter? What are the limitations? How is the topography calibrated? Please add data.

A3-6.

The characteristic feature of t-SPESI is that it can measure unevenness on the micrometer scale, which is difficult to do with AFM, with nanoscale precision by utilizing the oscillation of the capillary probe (Ref #55, Otsuka, Y. et al. High-spatial-resolution multimodal imaging by tapping-mode scanning probe electrospray ionization with feedback control. *Anal. Chem.* 93, 2263-2272 (2021)).

In this study, we used a step master to obtain the correlation between the height difference of 1-10 μm and the PID output signal (please refer to the answer to Q3-3). It was confirmed that the data for different heights could be fitted using a linear function (Supplementary Fig. 2b).

In addition, because the position resolution of the piezo stage, where the t-SPESI unit is fixed, is 1 nm and the repeat positioning accuracy is ± 1 nm, it was considered that the height of the sample in the range of up to 10 μm could be evaluated with nanoscale accuracy. Using the calibration data, the relative height of the topography was derived by converting the PID output voltage into the relative height.

One of the limitations of the measurement range of the feedback control is that it is not possible to measure samples with a height that exceeds the movable range of the piezo stage. In this case, since the maximum movement is 30 μm , it is necessary to use a larger piezo stage to capture height changes that exceed this.

<Added>

Page S-3 of the marked manuscript in the supplementary information.

(Captions in Supplementary Fig. 2b)

It was confirmed that the calibration data could be linearly fitted. In addition, the position resolution of the piezo stage for controlling the height of the probe was 1 nm, and the repeat positioning accuracy was ± 1 nm; therefore, the height of the sample in the range of up to 10 μm could be evaluated with nanoscale accuracy.

Q3-7.

Is there a way to get around shadowing during imaging?

A3-7.

(Regarding shadowing for ion images)

In t-SPESI, the probe was tilted 45 ° from the horizontal plane and fixed in place. It is necessary to make contact between the tip of the probe and the sample to perform extraction ionization. For samples in which the side of the probe makes contact with the sample surface (such as holes or convex shapes with a high aspect ratio), it may not be possible to make contact between the tip of the probe and the sample, and there may be areas where extraction-ionization cannot be performed. In the ion image obtained in this study, the probe was scanned from left to right in the horizontal direction, starting from the top-left corner of the image. If shadowing due to the sample shape occurs, extraction and ionization on the left side of the region with a large height (the cell nucleus) would not be possible, and a lack of ion signal intensity in the ion image would be observed. However, this did not occur, and ion signals were detected on both the left and right sides of the cell nucleus; therefore, it was thought that shadowing due to the sample shape did not occur.

(Regarding shadowing in amplitude images)

The dark areas in the amplitude image were due to a delay in the feedback control. As with AFM, this is an intrinsic cause of scanning probe microscopy. As shown in Fig. 4d and h, the amplitude increased and decreased owing to the delay in the feedback. The amplitude change was small compared to the amplitude value, so there was no effect that would cause extraction-ionization to disappear.

Q3-8.

What solvent is used for the measurements? Does the solvent matter?

A3-8.

DMF/MeOH was used as the solvent. A comparison between pure solvents (DMF and MeOH) and a mixed solvent showed that the mixed solvent could be used to measure lipid ions with high sensitivity (Ref #56, Y. Otsuka et al, Analyst 148, 1275-1284 (2023). <https://doi.org/10.1039/d2an01953a>). In addition, we reported that using this solvent suppresses the destruction of biological tissue and that it is possible to compare the cell distribution and lipid

distribution of the same tissue by performing H/E staining on the tissue after MSI (Ref #64, Y. Otsuka et al, *Anal. Bioanal. Chem.*, 417 275-286 (2025). <https://doi.org/10.1007/s00216-024-05641-x>).

The choice of the solvent requires further investigation. Because of the long measurement time in MSI, it is difficult to comprehensively investigate the combinations of candidate substances and solvents. It is necessary to develop a technology for high-throughput screening of extraction ionization using extraction solutions (mixtures containing solvents and additives such as standard substances and other cations) with different chemical compositions.

<Added>

Page 17, line 290 of the marked manuscript.

A mixture of DMF and MeOH was used to measure lipid ions (Ref #56). We previously reported that this mixture of solvents can suppress the disappearance of biological tissue, and that it is possible to perform hematoxylin and eosin staining of the tissue after MSI and compare the cell distribution and lipid distribution of the same tissue (Ref #64).

Q3-9.

How is the size of the liquid bridge measured? Please add data showing potential deviations from the stated 2.3 μm .

A3-9.

Thank you for your suggestion. To evaluate the spreading of the liquid bridges formed on the glass substrate, bright-field images were captured and analyzed. The average diameter was 4.0 μm , and the standard deviation was 0.87 μm . These results are larger than the diameters shown in Fig. 1c and show that the spreading of liquid bridges varies during measurement. The factors causing this variation are possibly due to fluctuations in the solvent flow and differences in the local wettability of the glass surface. The ion images presented here were measured using the oversampling method because the size of the liquid bridge was larger than the pixel size.

The oversampling method has been reported to be effective in improving the spatial resolution in MALDI-2 and Nano-DESI MSI (MALDI-2: J. C. McKinnon, R. Balez, R. S. E. Young, M. L. Brown, J. S. Lum, L. Robinson, M. E. Belov, L. Ooi, S. Tortorella, T. W. Mitchell, and S. R. Ellis, MALDI enabled oversampling for the mass spectrometry imaging of metabolites at single-cell resolution, *Journal of the American Society for Mass Spectrometry*, 35(11):2729–2742, 11 (2024). DOI: 10.1021/jasms.4c00241) (Nano-DESI: K. D. Duncan and I. Lanekoff, Oversampling to improve spatial resolution for liquid extraction mass spectrometry imaging, *Analytical Chemistry*, 90(4): 2451–2455, (2018). (DOI: 10.1021/acs.analchem.7b04687).

The main text has been revised, and supplementary figure and tables have been added.

<Removed>

Page 12, line 193 of the marked manuscript.

The diameter of the liquid bridge in Fig. 1c was approximately 2.3 μm when MSI was performed on HeLa cells.

Page 15, line 250 of the marked manuscript.

This volume was 125% greater than the volume of a hemisphere with a diameter of 1 μm .

<Added>

Page 12, line 194 of the marked manuscript.

To evaluate the spreading of the liquid bridges formed on the glass substrate, bright-field images were obtained and analyzed. The average diameter of the liquid bridges was 4.0 μm , with a standard deviation of 0.87 μm when MSI was performed on HeLa cells. (Supplementary Fig. 3 and Table 1). This result shows that the area of the liquid bridge varied during the measurements. The factors causing this variation are possibly the fluctuations in the solvent flow and differences in the local wettability of the glass surface. The ion images presented here were acquired using the oversampling method, because the size of the liquid bridge was larger than the pixel size.

Page 33, line 570 of the marked manuscript.

We obtained a snapshot of the bright-field image to evaluate the spreading of liquid bridges. On the glass substrate, the contrast of the liquid bridge changes. ImageJ was used to measure the area by enclosing the area with an ellipse. The diameter was calculated by assuming that the area was a perfect circle.

Page S-4 of the marked manuscript in the supplementary information.

Supplementary Fig.3.

Evaluation of spreading of the liquid bridge on the glass substrate during a single-line scan of the probe. Fig. 3a and l correspond to the snapshots at the beginning and end of the probe scan, respectively, and the others are snapshots taken during the scan, arranged in order of increasing time. The area of the liquid bridge is indicated by the yellow circles. Scale bar: 10 μm .

Page S-21 of the marked manuscript in the Supplementary Information.

Supplementary Table 1.

Analysis results for spreading area of liquid bridge. The types of images correspond to those shown in Fig. 3.

<Revised>

Page 15, line 251 of the marked manuscript.

Before: Considering that the diameter of the liquid bridges was 2.3 μm , approximately 8% of the solvent supplied during a single probe oscillation cycle

After: Considering that the diameter of the liquid bridges was 4 μm , approximately 67% of the solvent supplied during a single probe oscillation cycle

Page 48, line 902 of the marked manuscript.

Fig. 1c

The numerical values for the size of the liquid bridge have been deleted. The “Liquid bridge” was moved to the enlarged image. The enlarged area is indicated by a dashed square.

Q3-10.

How does the frequency correlate with the size of the liquid bridge? How does evaporation impact the measurements? Show data.

A3-10.

In this measurement, the frequency was set between 656 Hz and 657 Hz. Because the probe has a cantilever structure, the resonance frequency changes depending on the distance between the fixed part of the piezoelectric actuator, which provides the probe with oscillation energy and the oscillating probe tip.

The probe oscillation frequency is proportional to the number of extractions and ionizations per unit of time. Therefore, an increase in the oscillation frequency is thought to increase solvent consumption and reduce the volume of the liquid bridge. On the other hand, as you suggested, the time constants for the formation and rupture of the liquid bridge and electrospray ionization of the extracted solution may be affected by the physicochemical properties of the solvent (surface tension, viscosity, boiling point, etc.).

At present, it is not clear whether the size of liquid bridges can be reduced simply by increasing the oscillation frequencies. As shown in this study, we are at the stage of developing elemental technologies for single-cell MSI. We expect that it will be possible to optimize the oscillation conditions to realize MSI with higher spatial resolution by using the newly developed measurement system.

<Added>

Page 24, line 412 of the marked manuscript.

Optimization of the oscillation frequencies is important for improving the spatial resolution and increasing the ion signal intensity. To improve the spatial resolution, it is necessary to use a capillary probe with a finer tip and further reduce the size of the liquid bridge. However, as the solvent volume decreased, the number of extracted molecules decreased. One conceivable method to overcome this problem is to increase the oscillation frequencies for the extraction-ionization event, thereby preventing a decrease in the ion detection sensitivity. In t-SPESI, the resonance frequency of the capillary probe is proportional to the reciprocal of the square of the probe length (Ref #55). Therefore, the resonance frequency can be increased by reducing the probe length.

Q3-11.

How is the pixel size determined? How is it set? Please add data to validate the results.

A3-11.

In t-SPESI-MSI, the probe scanned the sample surface at a constant speed. The pixel size is defined by the distance the probe scans over a given time in the X-direction and the distance between the scanning lines in the Y-direction. Here, the X-direction corresponds to the scanning direction of the probe (fast scan axis), and the Y-direction corresponds to the direction orthogonal to the scanning direction of the probe (slow scan axis). This method is the same as other extraction-ionization methods (DESI, nano-DESI, LMJ-SSP, etc.). The pixel size (x) is expressed as $x = v * t$, where v is the scanning speed of the probe (v) and t is the time (t) taken to acquire a single mass spectrum.

In this measurement, $v = 6.7$ ($\mu\text{m/s}$) and $t = 300$ (msec), yielding a pixel size of $2 \mu\text{m}$.

The sample stage (BIOS-206T, OptoSigma) is closed-loop controlled and has a position resolution of $0.1 \mu\text{m}$, which is sufficient for positioning with a pixel size of $2 \mu\text{m}$. In addition, the minimum settable speed when moving at a constant stage speed is $0.1 \mu\text{m/sec}$, so there is no problem with the setting of $6.7 \mu\text{m/s}$.

<Added>

Page 32, line 548 of the marked manuscript.

The pixel size is defined by the distance the probe scans over a given time in the X-direction and the distance between the scanning lines in the Y-direction. Here, the X-direction corresponds to the scanning direction of the probe (fast scan axis), and the Y-direction corresponds to the direction orthogonal to the scanning direction of the probe (slow scan axis). In this study,

Q3-12.

The delay in feedback is stated to be minimal. Please add data to show this.

A3-12.

It is necessary to verify the section profile to determine if the feedback control works properly. The profiles shown in Fig. 4, i and j correspond to the results of scanning the probe from left to right.

In the height profile, an increase in height was observed at the location where the cells were present. Additionally, in the amplitude histogram, the values were within 1% of the amplitude values. Based on these two results, we thought that the feedback control worked properly.

We set the proportional and integral gains for the feedback control while checking the PID output signal and amplitude signal during the measurement. If the gain is not set appropriately and is too small, the probe's Z-axis position will not be adjusted in response to changes in the oscillation amplitude, and a profile that reflects the shape of the sample will not be obtained. If the gain is too high, oscillation of the PID output signal will occur, and the probe's Z-axis position will change more than the change in the oscillation amplitude of the probe, resulting in a distorted profile that does not reflect the shape of the sample. This adjustment method is common for scanning probe microscopy (t-SPESI and AFM). It is unusual to set an inappropriate gain value and acquire unnecessary data.

Q3-13.

Why is NaI used for optimization? And why is not the results from this used for the imaging of the cells? Including flow rate and temperature. Differences in applied voltage, use of from 300 nL/min to 1 nL/min, and 50 degrees difference in temperature suggests that the optimization is not used. Please optimize using metabolites in the same solvent and use the optimal settings for collecting the data.

A3-13.

As is clear from the MSI results, the signal intensity of the lipid ions changes depending on their position. Therefore, it is difficult to optimize the conditions of the ion transfer tube using the mass spectrum obtained from MSI of the cell.

Therefore, we investigated the relationship between temperature and ion signal intensity using NaI solution, which is commonly used as a calibration reagent for mass spectrometry.

Here, we used a syringe pump because we were concerned that using a nanopump would cause NaI to accumulate in the internal flow path, and that it would take time to clean the pump to replace the solvent for MSI (DMF/MeOH).

The solvent for the NaI solution was water/isopropanol, and its azeotropic boiling point was approximately 80 °C. (<https://pubs.acs.org/doi/pdf/10.1021/ja01438a004>) On the other hand, we were unable to find any reports of the azeotropic boiling point of DMF/MeOH. The boiling point of DMF at 1 atm (153 °C) is higher than that of water (100°C); therefore, the set temperature was increased by 50°C to achieve enhanced desolvation.

In this study, we showed that both the temperature and inner diameter of the ion transfer tube affect ion signal intensity when performing single-cell MSI using t-SPESI. We also demonstrated that we were able to successfully visualize lipid distribution within cells. The optimal conditions for each parameter were beyond the scope of this study.

As you suggested, we completely agree that the optimization of experimental parameters is necessary to improve the sensitivity of SC-MSI. For example, it is conceivable that optimization could be carried out using an extract of HeLa cells that had been homogenized and dissolved in DMF/MeOH. However, in phospholipid LC/MS research, it has also been shown that the appropriate temperature for desolvation differs depending on the class of phospholipid (Timothy J. Garrett, Matthew Merves, Richard A. Yost, Characterization of protonated phospholipids as fragile ions in quadrupole ion trap mass spectrometry, International Journal of Mass Spectrometry, 308, 299-306 (2011). <https://pmc.ncbi.nlm.nih.gov/articles/PMC3254096/>).

In MSI without chromatography, it is conceivable that optimization of the parameters of the ion transfer tube will vary depending on the target; therefore, it is necessary to select a standard sample according to the individual objective.

<Added>

Page 30, line 520 of the marked manuscript.

The boiling point of DMF (153°C) at 1 atm is higher than that of water (100°C). For the SC-MSI of HeLa cells, the temperature was set to 300°C to enhance desolvation. This value was 50°C higher than the temperature that showed the maximum value in the measurement of the NaI solution.

Q3-14.

The observed improvement using the smaller tube need to be confirmed with data. The speculations to why these observations occur need to be substantiated.

A3-14.

One side of the L-shaped ion transfer tube (the inlet of the gas flow) was at atmospheric pressure and located near the tip of the t-SPESI probe. The other side (the outlet of the gas flow) was located in front of the sampling cone of the mass spectrometer. The sampling cone and ion transfer tube were opposite to each other in the chamber and were isolated from the atmosphere. The sampling cone has a conical shape and has an aperture with a diameter of 0.67 mm at the tip. The interior of the sampling cone was evacuated using a turbomolecular pump in the mass spectrometer. Therefore, the vacuum level before the aperture is lower than that inside the ion transfer tube, and it is conceivable that the gas flow exiting the ion transfer tube expands spatially

before the sampling cone (Electrospray Ionization Mass Spectrometry: Fundamentals, Instrumentation, and Applications, edited by Ricard B. Cole, 1997).

To improve the transmission efficiency of ions generated at atmospheric pressure and fed into a mass spectrometer, it is necessary to increase the number of ions passing through the sampling cone aperture. It is conceivable that reducing the inner diameter of the ion transfer tube would reduce the directional dispersion of the gas flow from the outlet and increase the number of ions passing through the aperture by increasing the number of ions traveling in the axial direction of the transfer tube.

One of the purposes of this study was to show that it is possible to measure SC-MSI by increasing the ion signal intensity by changing the shape of the ion transfer tube. As you suggest, it is important to conduct research to prove this speculation to achieve further improvements in ion detection sensitivity. To achieve this, it is necessary to study the airflow inside the ion transfer tube in detail, so we would like to make this a challenge for the future. We have mentioned this in the main text.

Q3-15.

What is the patterned substrate used for the cells? In addition to the observataion, show data that this matters.

A3-15.

In this study, HeLa cells were randomly distributed because unmodified glass substrates were used. The technology for arranging cells in a regulated manner on patterned substrates is called single-cell patterning, and it has been used to obtain physical and biochemical information to understand cell behavior, such as cell morphology, differentiation, and apoptosis (R. Gayathri et al., *Materials Today Chemistry* 26 (2022) 101021, <https://doi.org/10.1016/j.mtchem.2022.101021>).

Combining single-cell patterning with SC-MSI is expected to be effective for profiling the metabolites of cells cultured under more standardized conditions and their molecular-level responses to stimuli such as drug administration.

<Added>

Page 26, line 433 of the marked manuscript.

Techniques for arranging cells in a regular pattern on a patterned substrate have been used to obtain physical and biochemical information to understand cell behavior, such as cell morphology, differentiation, and apoptosis (Ref #74). By combining these technologies with SC-MSI, cell metabolites and responses to drug administration can be profiled at the molecular level under well-defined conditions.

Q3-16.

How is the chemical and spatial integrity of the cells preserved? What does the staining and fixation do to the mass spectrum and contaminations? Are the contaminations close to the m/z of the reported lipids? Please show data showing mass spectrum of only cells, only staining, only fixation and the combination of the three.

A3-16.

The main purpose of this paper is to present a demonstration of multimodal imaging of the same single cell using fluorescence microscopy and t-SPESI.

When no pretreatment is performed, that is, when cells cultured on a glass substrate in culture medium are dried under atmospheric conditions, the high concentrations of salts and other solutes in the culture medium crystallize to cover the glass substrate and cells, making it difficult to observe the distribution of cells clearly. These salts are preferentially ionized in electrospray ionization; therefore, it is conceivable that they inhibit the detection of ions derived from cells. Therefore, we did not measure the cells as they were cultured, as this was not suitable for the purpose of this study.

Next, in the pretreatment to remove the culture medium, it is known that if the cells are rinsed with a low-salt aqueous solution, the balance of osmotic pressure is disrupted, causing the cells to swell and the cell membrane to break. Therefore, the cells were chemically fixed according to Ref #76. This reference describes the effectiveness of chemical fixation and reports that fixation with glutaraldehyde and rinsing with ammonium acetate can maintain cell morphology and enable lipid detection. Chemical fixation with glutaraldehyde is important for detecting phospholipids in SC-MSI using MALDI. Therefore, it is conceivable that chemical fixation is important for maintaining cell morphology and detecting lipids in SC-MSI. (Yvonne Schober et al., *Anal. Chem.* 2012, 84, 15, 6293–6297, <https://doi.org/10.1021/ac301337h>).

As shown in this paper, it was confirmed that chemical fixation with glutaraldehyde and rinsing with ammonium acetate did not cause any significant problems in the results of staining with fluorescent dyes (the nucleus and cytoplasm were stained, respectively). However, since the ions of the fluorescent dyes Hoechst 33342 and fluorescein diacetate were not detected, it was assumed that these two fluorescent dyes did not have an ionization suppression effect on the ionization of phospholipids.

In addition, while SC-MSI can be performed on chemically fixed, unstained cells, it is difficult to determine the positions of both the nucleus and the cell because the only way to identify their positions is by contrast in the brightfield image. Therefore, to demonstrate multimodal imaging using the measurement system we developed, SC-MSI on unstained cells is beyond the scope of this study.

On the other hand, in recent years, multimodal imaging combining MSI and fluorescence microscopy has attracted attention. In particular, in the case of SC-MSI of biological tissues, the challenge of the measurement time increases nonlinearly as the pixel size decreases. To reduce the measurement time, one practical solution could be to limit the area to be measured using a fluorescence microscope, which enables faster observation and then perform MSI. It is conceivable that, in the future, it will be necessary to verify that fluorescent molecules do not act as inhibitory factors for the ionization of target molecules in MSI.

Q3-17.

How do you know that the cells are not overlapping on the surface?

A3-17.

HeLa cells attach to a substrate and proliferate. When the cell density was low, HeLa cells spread laterally as far as there was space, so they almost never overlapped. However, when the cells proliferate and reach a confluent state, they come into contact with each other, and some begin to overlap.

In this study, we used HeLa cells in a state before they reached confluence, by adjusting the cell culture time. We confirmed that there was space on the glass substrate and that HeLa cells were present.

If the HeLa cells overlapped on the glass substrate, it would be conceivable that there would be an area where the cytoplasm of a certain HeLa cell overlapped with the nucleus of another HeLa cell, and the contrast of the fluorescent image would change (green and blue would overlap), but such a result was not confirmed in the sample used in this study.

<Added>

Page 26, line 446 of the marked manuscript.

The cell culture time was adjusted to use HeLa cells in a preconfluent state. By observing the cells using a fluorescence microscope, it was confirmed that there was no overlap between the adjacent HeLa cells.

Q3-18.

It does not seem like the ROI is of 45 cells since less than this are shown in the figures.

A3-18.

The results of SC-MSI for other HeLa cells have been added to Supplementary Fig. 7 and an explanation has been added to the main text.

<Added>

Page 21, line 360 of the marked manuscript.

The other ion images of HeLa cells are shown in Supplementary Fig. 8.

<Added>

Supplementary Fig. 8

Q3-19.

Please add the mass resolving power of the mass spectrometer used to allow for a mass tolerance of 0.005 Da. Only high res qtof can provide data with the three decimals used in the manuscript. Annotation on low res MS data is not recommended.

A3-19.

When evaluating mass spectra, it is important to consider both mass resolving power and mass accuracy. The mass resolving power of the Q-TOF used was >30,000 FWHM in the sensitivity mode. In addition, the mass accuracy when lock mass was applied was 1 ppm.

(https://lcms.labrulez.com/labrulez-bucket-strapih3hsga3/720005077_EN_ae96fd30be/720005077EN.pdf)

All the data obtained were corrected for m/z using the ion peaks at m/z 760.5851 (PC 34:1 [M+H]⁺) and m/z 734.5694 (PC 32:0 [M+H]⁺). The error from the theoretical value for each ion peak was 0.2 ppm for both. This value was within the mass accuracy range of the Q-TOF.

The mass accuracy affects the search in the LIPID MAPS database. When the mass accuracy is 1 ppm, the error range for the ion peak at m/z 700 is ±0.0007 Da. Therefore, it is conceivable that a mass tolerance of ±0.005 is appropriate for the LIPID MAPS database search for the data analysis in this study.

In addition, mass resolving power affects the separation of adjacent ion peaks. Considering the mass resolving power, the Δm of ion peaks in the m/z 700-900 range is within the range of 0.02-0.03. Considering this error, all the ion peaks in the current study are outside the Δm range, so it can be said that the ion peaks were separated.

<Added>

Page 32, line 547 of the marked manuscript.

The mass resolving power of the mass spectrometer was >30000 FWHM in sensitivity mode.

Q3-20.

How was the significance calculated between the two cell types? P-values? Please specify.

(The same as the answer to the second question from Reviewer 2.)

A3-20.

Welch's t-test for unequal variances was performed using Microsoft Excel for data analysis. As we had not verified whether the actual data had unequal variance, we performed Levene's test, as suggested. As a result, among the data for the 29 ions considered in this study, the p-value for the 17 ions fell below the significance level (0.05). The data for the 12 ions that did not meet the significance level were considered to have no difference in variance, and a reanalysis of Welch's t-test for two samples with equal variance was performed. Consequently, the p-value of the t-test changed, but the change was minimal, ranging from 0 to 1.3E-5. In Supplementary Table 5, a column showing the results of Levene's test has been added, and the results of the reanalysis of Welch's t-test have been revised.

<Added>

Page 34, line 586 of the marked manuscript.

Levene's test and Welch's t-test were used to assess significant differences in lipid ion signal intensities between the cell types. The variance between the two groups was examined using Levene's test for signal intensity data of the ROIs. Among the data for the 29 ions, the p-values for the 17 ions were below the significance level (0.05). The data for the 17 ions were analyzed using Welch's t-test for two samples with unequal variances. The data for the 12 ions were analyzed using Welch's t-test for two samples with equal variances.

Q3-21.

How many endogenous molecules was detected from the cells? The manuscript only states small amount.

A3-21.

As described in the section on "Distinguishing cells by lipid information," we selected 166 ion peaks by searching the Lipidmaps database using the measured mass spectra. We also performed a quantitative analysis using SFC-MS/MS. Of the 334 lipids confirmed to be present, 47 were tentatively assigned using t-SPESI-MSI. A t-test was performed on the ion images of the 47 lipid ions, and the number was narrowed to 29 ions.

As a supplement, more ion peaks were observed in the acquired mass spectra than those mentioned above. We expect that more lipid ions will be assigned when ion mobility mass spectrometry and MS/MS analyses are available.

Q3-22.

The meaning of the distribution of lipids in the cells seem highly overstated. There are no lipids inside the nucleus, which is clear in the PC images. This makes me think that the other images show contaminations and noise and not distributions of endogenous lipids.

A3-22.

The PC ion image clearly showed a decrease in signal intensity in the nuclear region. This indicated that the amount of lipids in the nucleus was lower than that in other regions. Notably, the signal intensity of lipids in the nuclear region was not zero.

In t-SPESI, the solvent penetrates from the surface to the interior of the cell, causing the extraction and ionization of the components. (Answer to Reviewer 2's question 2. See Supplementary Fig. 11). Therefore, it is conceivable that each pixel of the ion image of the cells indicates the ionization of lipids, both on the cell surface and inside the cell. Therefore, even in the region where the nucleus is present, the signal intensity of PC does not become zero because the cell membrane, which is located higher than the nucleus, is ionized. The ion images of lipids assigned to SM and HexCer were characterized by the fact that the signal intensity of ions was higher in the region where the cell was present than in the glass substrate and that the relative signal intensity differed depending on the cell type. If there was contamination (chemical noise) due to experimental procedures or sample preparation, or electrical noise intrinsic to the measurement system, it would be conceivable that ions would be detected throughout the sample, or that the signal intensity would be similar between different cell types. However, this was not the case in the experimental results.

Q3-23.

The experiment to try to correlate quantification with SFC and t-SPESI is not understandable and needs to be revised.

A3-23.

In this study, we referred to the results of SFC-MS/MS to support the estimation of ion peaks measured by t-SPESI. It is commonly recognized that quantitative evaluation of MSI is challenging because of ionization suppression by coexisting substances in the sample. As shown in Supplementary Fig. 10, it was actually recognized that there was no correlation between the results of quantitative analysis by SFC-MS/MS and the ion signal intensities of t-SPESI-MSI.

We believe that the results showing the difficulty of quantitative analysis using t-SPESI-MSI are important for future research aimed at quantitative evaluation. This is because the correlation analysis approach shown in this paper is important when examining the effects of the physicochemical properties of the solvent and the internal standard added to the solvent on the ion signal intensity.

Q3-24.

How was the scanning speed of the probe optimized to the very specific 6.7 $\mu\text{m/s}$? Please show data.

A3-11.

The answer is the same as that given in A3-11; therefore, it has been omitted.

Other revisions

1. The numbers have been revised to match the addition of the figures, tables, and references.
2. The parentheses are removed from the figure labels.
3. The notation of the ions in the ion image has been corrected in Fig. 4.

Before: $[M:H]^+$

After: $[M+H]^+$

4. Added Acknowledgements
The Asahi Glass Foundation

That's all. Thank you very much.